**PLOS** NEGLECTED TROPICAL DISEASES

# On-target inhibition of *Cryptosporidium parvum* by nitazoxanide (NTZ) and paclitaxel (PTX) validated using a novel *MDR1*-transgenic host cell model and algorithms to quantify the effect on the parasite target

Bo Yang[☯], Yueyang Yan[☯], Dongqiang Wang, Ying Zhang, Jigang Yin, Guan Zhu[ID]*

State Key Laboratory for Zoonotic Diseases, Key Laboratory of Zoonosis Research of the Ministry of Education, the Institute of Zoonosis, and the College of Veterinary Medicine, Jilin University, Changchun, China

☯ These authors contributed equally to this work.
* cryptosporida@gmail.com

## Abstract

*Cryptosporidium parvum* is a globally distributed zoonotic protozoan parasite that causes moderate to severe, sometime deadly, watery diarrhea in humans and animals, for which fully effective treatments are yet unavailable. In studying the mechanism of action of drugs against intracellular pathogens, it is important to validate whether the observed anti-infective activity is attributed to the drug action on the pathogen or host target. For the epicellular parasite *Cryptosporidium*, we have previously developed a concept that the host cells with significantly increased drug tolerance by transient overexpression of the multidrug resistance protein-1 (MDR1) could be utilized to evaluate whether and how much the observed anti-cryptosporidial activity of an inhibitor was attributed to the inhibitor's action on the parasite target. However, the transient transfection model was only applicable to evaluating native MDR1 substrates. Here we report an advanced model using stable MDR1-transgenic HCT-8 cells that allows rapid development of novel resistance to non-MDR1 substrates by multiple rounds of drug selection. Using the new model, we successfully validated that nitazoxanide, a non-MDR1 substrate and the only FDA-approved drug to treat human cryptosporidiosis, killed *C. parvum* by fully (100%) acting on the parasite target. We also confirmed that paclitaxel acted fully on the parasite target, while several other inhibitors including mitoxantrone, doxorubicin, vincristine and ivermectin acted partially on the parasite targets. Additionally, we developed mathematical models to quantify the proportional contribution of the on-parasite-target effect to the observed anti-cryptosporidial activity and to evaluate the relationships between several in vitro parameters, including antiparasitic efficacy ($EC_i$), cytotoxicity ($TC_i$), selectivity index ($SI$) and Hill slope ($h$). Owning to the promiscuity of the MDR1 efflux pump, the *MDR1*-transgenic host cell model could be applied to assess the on-parasite-target effects of newly identified hits/leads, either substrates or non-substrates of MDR1, against *Cryptosporidium* or other epicellular pathogens.

**Data Availability Statement:** All relevant data are within the manuscript and its Supporting Information files.

**Funding:** This research was funded in part by the National Natural Science Foundation of China (grant number 32250710141 to GZ) and National Key R&D Program of China (grant number 2017YFC1601200 to JY). The funders had no role in study design, data collection and analysis, decision to publish, or preparation of the manuscript.

**Competing interests:** The authors have declared that no competing interests exist.

## Author summary

*Cryptosporidium parvum* is an important zoonotic parasite, for which fully effective treatments are unavailable. Anti-cryptosporidial drug discovery faces many challenges and technical difficulties. One obstacle is the lack of tools to assess whether the killing of *C. parvum* by an inhibitor is attributed to the action on the parasite or on host cells. To address this question, we developed an *MDR1*-transgenic host cell line that allowed rapid development of drug resistance by applying continuous drug pressure. By analyzing the antiparasitic activity and cytotoxicity between wild-type and drug-resistant host cells, we verified that nitazoxanide (the only FDA-approved drug for treating cryptosporidiosis) and paclitaxel (anti-cryptosporidial lead) killed the parasite by acting fully on the parasite, whereas mitoxantrone, doxorubicin, vincristine and ivermectin killed the parasite by acting on both the parasite and host cells. We also developed algorithms to differentiate the percent contributions of actions on the parasite and host cell targets to the observed anti-cryptosporidial activity. In summary, we developed novel in vitro and mathematical models for evaluating the on-target effects of anti-cryptosporidial drugs. The models are also applicable to evaluate/quantify the drug actions on other epicellular pathogens.

## Introduction

Cryptosporidiosis is a globally distributed diarrheal disease of humans and animals. Among more than 40 *Cryptosporidium* species or genotypes, humans are mainly infected by *C. parvum* (zoonotic) and *C. hominis* (anthroponotic), while immunocompromised patients might also be infected by other species [1–3]. In people with weak or compromised immunity (e.g., infants, elderly and AIDS patients), cryptosporidial infection can be severe or deadly. Cryptosporidiosis is also a significant problem in farm animals and may cause death in neonatal calves and substantial weight loss in cattle [4,5]. On the other hand, only a single drug (i.e., nitazoxanide [NTZ]) is approved by the United States Food and Drug Administration (FDA) for treating human cryptosporidiosis. However, NTZ is not fully effective in immunocompetent patients and ineffective in immunocompromised individuals, and its mechanism of action remains undefined [6,7].

While the anti-cryptosporidial drug discovery has been impeded by some technical constraints (e.g., difficulties in manipulating the parasite in vitro and in vivo) and unique parasite biology (e.g., lack of conventional drug targets and epicellular parasitic lifestyle), an increasing effort in the past decade has resulted in the discovery of a number of leads showing excellent anti-cryptosporidial efficacy in vitro and in animal models [8–13]. Hits or leads might be identified by in vitro phenotypic screening or by target-based screening, followed by confirmation of efficacy in vitro and in vivo. For obligate intracellular parasites including *Cryptosporidium*, an efficacious drug may kill the parasite directly via acting on a parasite target or indirectly via acting on a host cell target, or both (i.e., the actions on both the parasite and host targets contributing to the killing of the parasite) (see illustration in S1 Fig). For simplicity, hereinafter we will use "on-target" effect to describe the action of an inhibitor "on the parasite target" and "off-target" effect to describe the action of an inhibitor "off the parasite target" (i.e., on the host target).

The validation and quantification on whether and how much a hit/lead truly inhibits the parasite by acting on the parasite target is technically challenging for *Cryptosporidium* and other obligate intracellular pathogens. There were actually few attempts to demonstrate on-

target effect of anti-cryptosporidial leads, mainly by analyzing coefficients between the inhibitory activities of hit/lead analogs on a defined target ($K_i$ or $IC_{50}$ values) and their in vitro anti-cryptosporidial efficacies ($EC_{50}$ values), e.g., the actions of inhibitors of phosphatidylinositol-4-OH kinase [PI(4)K] and methionyl tRNA-synthetase (MetRS) [10,14]. Among the anti-cryptosporidial leads discovered in the past decades, there is a general lack of experimental evidence to differentiate the contributions of actions on the parasite target from those on the host target to the observed efficacy. Even for NTZ and paromomycin, the two classic anti-cryptosporidial compounds and standards, there is a need to understand the mechanisms of action (MOA), which still remains speculative, but is important in optimization of drugs, reduction in toxic effects and identification of new leads with the same or similar MOA.

We have recently developed an HCT-8 cell-based transient *MDR1*-transfection model for evaluating the routes of drug actions on *C. parvum* [15]. Human MDR1 (multidrug resistance protein 1), also known as P-gp (P-glycoprotein 1) or ABCB1 (ATP-binding cassette subfamily B member 1), is an ATP-dependent efflux pump with broad substrate specificity [16,17]. The model takes advantage of the substrate promiscuity of MDR1 for rapid development of drug resistance in host cells on selected anti-cryptosporidial compounds. Because *C. parvum* is an epicellular parasite that is consistently exposed to the drug present in the culture medium (Fig 1), the on-target effects can be evaluated by determining whether the MDR1-mediated increase of drug resistance in the host cells affects the anti-cryptosporidial efficacy of the drug. If a drug inhibits the parasite growth by solely acting on the parasite (100% on-target), the change of drug tolerance in the host cells would not affect the anti-cryptosporidial efficacy of the drug. Using this model, we have confirmed that the previously discovered lead paclitaxel (PTX) inhibits the growth of *C. parvum* solely by its action on the parasite (i.e., on-target effect fully contributed to the killing of the parasite), while several other compounds act on both the parasite and host cell targets (i.e., both on- and off-target effects contributed to the killing of the parasite).

The transient *MDR1*-transfection model can be quickly established with increased tolerance to multiple drugs that are native MDR1 substrates (e.g., paclitaxel). However, it is inapplicable to non-substrates of MDR1 (e.g., NTZ) [15]. Here we report the development of a new model with the potential to evaluate anti-cryptosporidial on-target effects of unrestricted classes of compounds. Based on the ligand promiscuity of MDR1 [18,19], we hypothesize that cells over-expressing MDR1 would be more adaptable to developing resistance to xenobiotics. Therefore, drug resistance would be more rapidly developed to both substrates and non-substrates of MDR1 in transgenic cells that stably overexpress *MDR1* and receive continuous drug pressure. To test this hypothesis, we established a transgenic HCT-8 cell line stably transfected with *MDR1*, applied drug pressures to the transgenic cells with an MDR1 substrate paclitaxel (PTX) and a non-MDR1 substrate nitazoxanide (NTZ), and successfully obtained two cell lines with significant increase of resistance to PTX (>3-fold increase over negative control) and to NTZ (>2-fold increase) in three months. The PTX-resistant cells also displayed increased resistance to several other compounds, including ivermectin (IVM), vincristine (VCT), doxorubicin (DRB) and mitoxantrone (MXT).

Using these cell lines, we validated that NTZ and PTX inhibited the growth of *C. parvum* in vitro by fully acting on the parasite target (i.e., 100% on-target), while the inhibition by IVM, VCT, DRB and MXT were attributed to their actions on both the parasite and host cell targets (i.e., partially on-target). This is the first time that the on-target effect was confirmed for the only FDA-approved anti-cryptosporidial drug NTZ. Additionally, we developed algorithms to quantify the theoretical proportions of contribution of on-target and off-target effects of compounds to the observed anti-cryptosporidial activity in vitro.

**Fig 1. Effect of MDR1 overexpression in host cells on the exposure of an MDR1 substrate to the parasite and host targets in *C. parvum* in vitro system.** The epicellular *C. parvum* embraced by a parasitophorous vacuole membrane (PVM) is directly exposed to the culture medium, but separated from the host cell cytoplasm by an electron-dense structure. A vertical line is drawn to compare the drug exposure between the parasite grown on a wild-type (WT) or negative control (NC) host cell (on the left) and that on a transgenic cell overexpressing *MDR1* (on the right). Blue dots illustrate a hypothetical compound under investigation. Overexpression of *MDR1* gene in the host cell would not affect the concentration of the compound in the culture medium (i.e., [a] = [b]). In WT/NC cells, there is a slight decrease of compound concentration in the cytoplasm due to the basal level of MDR1-mediated efflux (i.e., [a] = [b] > [c]). In *MDR1*-overexpressing cells, there is a greater decrease of compound concentration in the cytoplasm due to the higher level of MDR1-mediated efflux (i.e., [a] = [b] > [c] >> [d]). The presence of a basal level of MDR1 also explains why the MDR1 inhibitor elacridar increased the cytotoxicity of some compounds to WT/NC cells as shown in Figs 8 and 9.

## Results

### Transgenic HCT-8 cells overexpressing *MDR1* allow relatively rapid development of drug resistance to both substrates and non-substrates of MDR1

We first generated a stable transgenic cell line by transfection of HCT-8 cells with a lentiviral vector carrying copepod *GFP* (*copGFP*) and human *MDR1* genes driven by EF1α and CMV

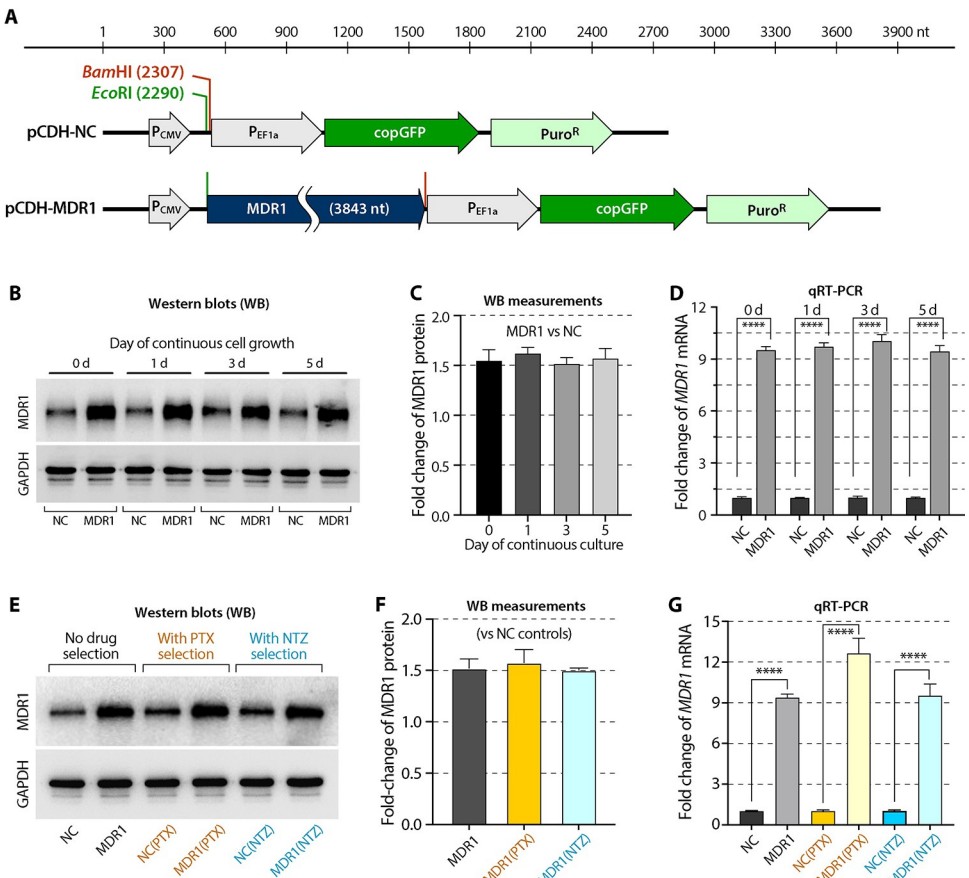

**Fig 2. Vectors for generating MDR1-transgenic HCT-8 cells and confirmation of the MDR1 overexpression in _MDR1_-transgenic cells.** (A) Illustration of the negative control blank vector (pCDH-NC) containing a copGFP driven by _EF1α_ promoter and its derived vector carrying the whole _MDR1_ open reading frame (cDNA reverse-transcribed from mRNA) driven by CMV promoter (pCDH-MDR1). _Eco_RI and _Bam_HI refer to the restriction sites for the insertion of _MDR1_ cDNA fragment. (B) Western blots of MDR1 protein in HCT-8/NC and HCT-8/MDR1 cells (labeled as NC and MDR1, respectively). GAPDH was used as an interval control. A set of representative blots is shown here. (C) Fold change of MDR1 protein levels between MDR1 and NC cells as measured from the western blots and normalized with GAPDH ($n = 3$). (D) Fold change of _MDR1_ mRNA between MDR1 and NC cells as determined by qRT-PCR and normalized with _GAPDH_ ($n = 3$). In panels B, C and D, cells were continuous cultured for 0, 1, 3 and 5 days to confirm the consistency of _MDR1_-overexpression in _MDR1_-transgenic cells. (E) Western blots of MDR1 protein in NC and MDR1 cells in comparison with those after drug selections by paclitaxel (labeled as NC(PTX) and MDR1(PTX)) or nitazoxanide (labeled as NC(NTZ) and MDR1(NTZ)) ($n = 3$). (F) Fold change of MDR1 protein in _MDR1_-overexpressing cells in comparison with corresponding NC cells [i.e., MDR1 vs. NC, MDR1(PTX) vs. NC (PTX) and MDR1(NTZ) vs. NC(NTZ)] as measured from the western blots and normalized with GAPDH ($n = 3$). (G) Fold change of _MDR1_ mRNA in _MDR1_-overexpressing cells in comparison with corresponding NC cells [i.e., MDR1 vs. NC, MDR1(PTX) vs. NC(PTX) and MDR1(NTZ) vs. NC(NTZ)] as determined by qRT-PCR and normalized with GAPDH ($n = 3$). Panels E, F and G show that drug selections by PTX and NTZ had no or little effect on the expression of _MDR1_ at both mRNA and protein levels. Bars represent the standard errors of the means (SEMs; $n = 3$). Statistical significances were determined by Holm-Šídák multiple _t_-test between group pairs (**** = $p < 0.0001$).

promoters, respectively (Fig 2A). Parental HCT-8 cells (wild-type) and those carrying blank vectors (negative control) or _MDR1_ gene were designated as HCT-8/WT, HCT-8/NC or HCT-8/MDR1 cells, or WT, NC or MDR1 cells for simplicity (Table 1). MDR1 cells continuously overexpressed MDR1 as demonstrated at both protein and mRNA levels (Fig 2B–2D). In comparison to NC cells, there were >1.5-fold increase of MDR1 protein and >9-fold increases of mRNA in MDR1 cells, respectively (Fig 2C and 2D). The fold increases were lower, but the

**Table 1. List of cell lines used in this study.**

| Cell lines | Abbreviations | Drug selection | Description |
|---|---|---|---|
| HCT-8/WT | WT | None | Parental HCT-8 cells |
| HCT-8/NC | NC | None | Transgenic HCT-8 cells with blank vector (containing *copGFP* gene; negative control) |
| HCT-8/MDR1 | MDR1 | None | Transgenic HCT-8 cells over-expressing *MDR1* and *copGFP* genes |
| HCT-8/WT(PTX) | WT(PTX) | Paclitaxel (PTX) | HCT-8/WT cells after drug selection pressure by paclitaxel (PTX) |
| HCT-8/NC(PTX) | NC(PTX) | Paclitaxel (PTX) | HCT-8/NC cells after drug selection pressure by paclitaxel (PTX) |
| HCT-8/MDR1(PTX) | MDR1(PTX) | Paclitaxel (PTX) | HCT-8/MDR1 cells after drug selection pressure by paclitaxel (PTX) |
| HCT-8/WT(NTZ) | WT(NTZ) | Nitazoxanide (NTZ) | HCT-8/WT cells after drug selection pressure by nitazoxanide (NTZ) |
| HCT-8/NC(NTZ) | NC(NTZ) | Nitazoxanide (NTZ) | HCT-8/NC cells after drug selection pressure by nitazoxanide (NTZ) |
| HCT-8/MDR1(NTZ) | MDR1(NTZ) | Nitazoxanide (NTZ) | HCT-8/MDR1 cells after drug selection pressure nitazoxanide (NTZ) |

levels were more consistent over the time, than those in our previously reported transiently transfected cells (i.e., 2.12- to 3.37-fold for protein and >40-fold for mRNA) [15].

The transgenic cell lines were evaluated for their drug tolerance to nine compounds by MTS cytotoxicity assay, including the anti-cryptosporidial lead PTX and the only approved drug NTZ (see the description of compounds tested in this study in S1 Table). Based on the 50% cytotoxic concentrations ($TC_{50}$), MDR1 cells showed 1.63-fold increase of tolerance to PTX ($TC_{50}$ = 20.22 μM; vs. 12.37 in NC or 12.75 μM in WT cells) (Fig 3A; Table 2, left four columns), but no change of tolerance to NTZ (i.e., $TC_{50}$ = 25.28, 25.90 and 26.75 μM on the three cell lines) (Table 2; Fig 3B). The results agreed with the fact that PTX was a native substrate of MDR1, whereas NTZ was not [19–21]. The 1.63-fold increase of tolerance to PTX in the MDR1 cells was lower than the >2-fold increase in transiently transfected cells as previously reported [15]. MDR1 cells (vs. NC or WT cells) also exhibited increased tolerance to four of the other seven compounds, i.e., 1.54- to 1.76-fold increases to mitoxantrone (MTX), doxorubicin (DXR), vincristine (VCT) and ivermectin (IVM), but not to cyclosporin A (CSA), daunorubicin (DRC) and loperamide (LPM) (i.e., 0.95- to 1.03-fold changes) (Table 2).

To test the hypothesis that *MDR1*-transgenic cells were more adaptable to drug selection for developing drug resistance to the "substrates of MDR1", we applied continuous drug pressures with stepwise increase of concentrations of PTX to MDR1 cells (vs. WT and NC cells; drug selection design in S2 Table). For clarity, a cell line after drug selection was named by adding abbreviation of the drug in parenthesis, e.g., WT(PTX), NC(PTX) or MDR1(PTX) (Table 1). After selection with PTX, all three resulting cell lines [i.e., WT(PTX), NC(PTX) and MDR1(PTX)] increased tolerance to PTX. For comparison of cells before and after PTX selection, WT(PTX) and NC(PTX) cells showed smaller increases of PTX-resistance (i.e., 1.26- or 1.23-fold increase of $TC_{50}$ vs. WT or NC cells) (Fig 3C; Table 2, middle four columns), while MDR1(PTX) cells showed a much larger increase of PTX-resistance (i.e., 2.27-fold increase vs. MDR1 cells) (Table 2, right four columns). These observations confirmed that cells overexpressing *MDR1* could develop greater drug resistance under drug pressure than WT and NC cells. In comparison to the negative control cells [e.g., NC(PTX) and NC cells], MDR1(PTX) cells displayed much higher resistance to PTX (i.e., 3.02- and 3.71-fold increases, respectively). Cells after PTX selection also increased resistance to five other compounds, including MTX, DXR, VCT, IVM and DRC (2.48- to 2.95-fold vs NC(PTX); or 2.38- to 3.01-fold vs NC cells), but not to CSA, LPM and NTZ (Table 2; Fig 3D). It was notable that the resistance to DRC was successfully increased in MDR1(PTX) cells (2.46-fold vs. NC cells) that was unachieved in MDR1 cells (0.95-fold vs. NC cells).

We also tested the hypothesis that drug resistance to "non-substrates of MDR1" could be rapidly developed in *MDR1*-transgenic cells. NTZ was chosen here because it was the only

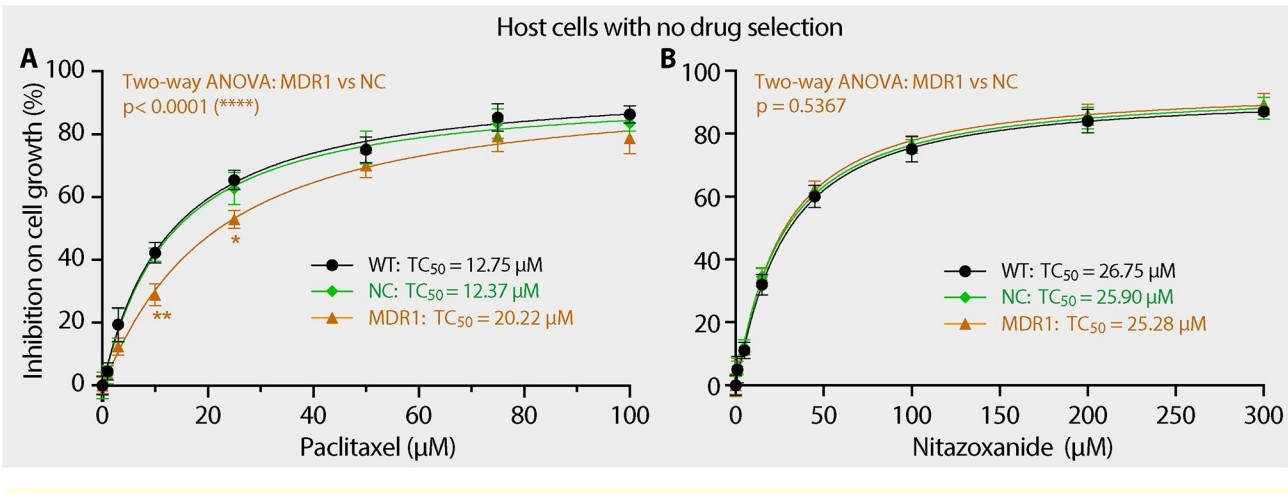

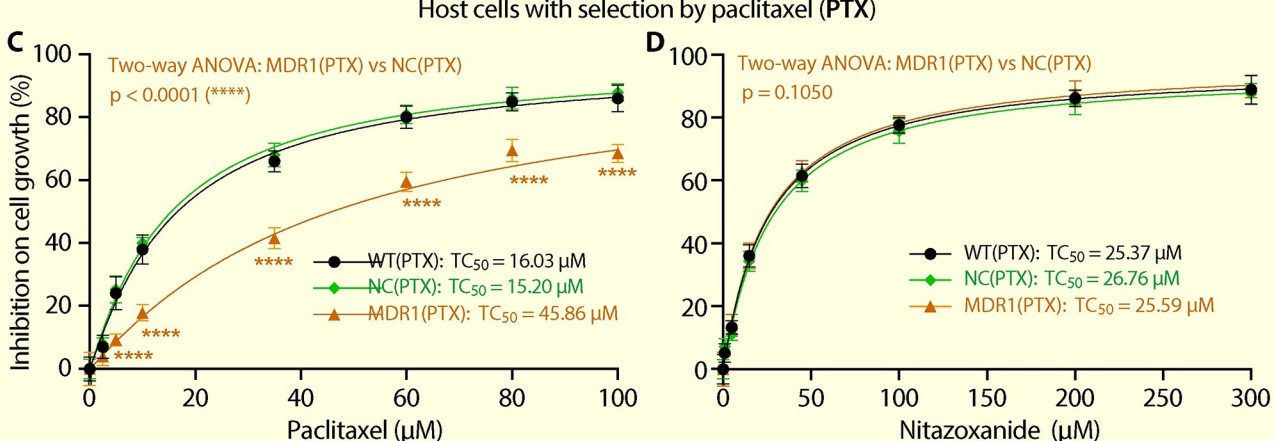

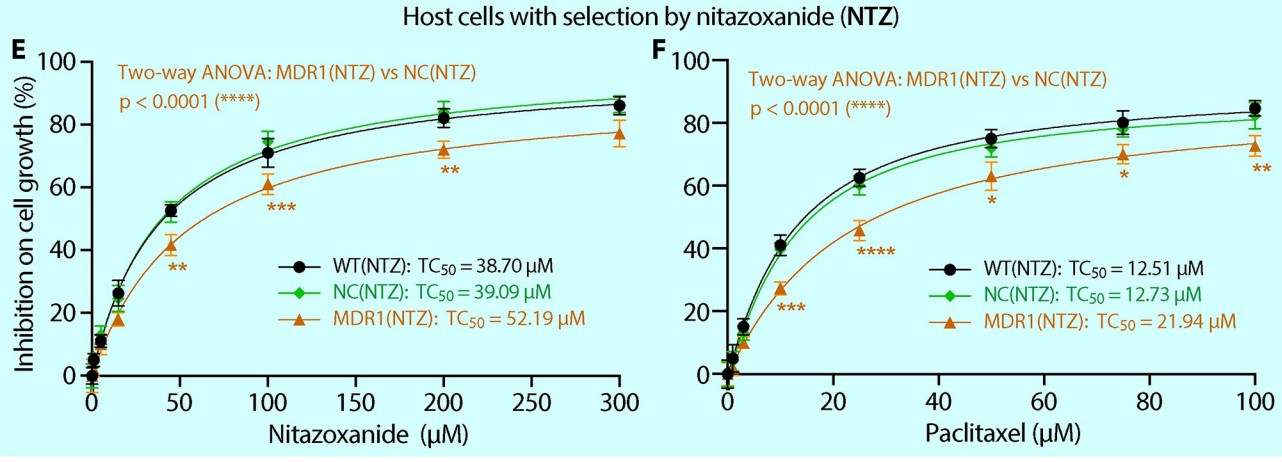

**Fig 3. Effects overexpressing MDR1 and drug selection by paclitaxel (PTX) or nitazoxanide (NTZ) on the tolerance of cells to PTX or NTZ as determined by MTS cytotoxicity assay.** (A, B) Inhibition of PTX (A) or NTZ (B) on the growth of the wild-type, blank vector negative control and MDR1-transgenic HCT-8 cells (labeled as WT, NC and MDR1, respectively). (C, D) Inhibition of PTX (C) or NTZ (D) on the growth of WT, NC and MDR1 cells that were subjected to the selection by PTX [labeled as WT(PTX), NC(PTX) and MDR1(PTX), respectively]. (E, F) Inhibition of PTX (F) or NTZ (E) on the growth of WT, NC and MDR1 cells subjected to the selection by NTZ [labeled as WT(NTZ), NC(NTZ) and MDR1(NTZ), respectively]. Also see Table 1 for more detailed descriptions about the cell lines. $TC_{50}$ = 50% cytotoxicity values. Bars represent the standard errors of the means (SEMs; $n$ = 3). Statistical significances between curves were determined by two-way ANOVA with $p$-values shown in brown fonts. Statistical significances between individual data points (i.e., MDR1, MDR1(PTX) or MDR1(NTZ) cells vs. corresponding NC, NC(PTX) and NC(NTZ) cells, respectively) were determined by Holm-Šídák multiple $t$-test between group pairs (* = $p$ <0.05, ** = $p$ <0.01, *** = $p$ <0.001 and **** = $p$ <0.0001). There are no statistical significances between WT and NC, WT(PTX) and NC(PTX), as well as WT(NTZ) and NC(NTZ) cells in both two-way ANONA and multiple $t$-tests.

**Table 2. Drug tolerance profiles of the three host cell lines before and after selection by paclitaxel (PTX) as determined by MTS assay and expressed in 50% inhibition concentrations ($TC_{50}$ values).**

| Compounds | $TC_{50}$ in cells without drug selection (μM) and fold changes (vs NC) | | | | $TC_{50}$ in cells with selection by PTX (μM) and fold changes (vs NC(PTX)) | | | | Fold changes of $TC_{50}$ after selection by PTX (vs their parent cell lines) | | | |
|---|---|---|---|---|---|---|---|---|---|---|---|---|
| | WT | NC | MDR1 | MDR1/NC* | WT (PTX) | NC (PTX) | MDR1 (PTX) | MDR1(PTX)/NC(PTX)* | WT(PTX) vs. WT | NC(PTX) vs. NC | MDR1(PTX) vs. MDR1 | MDR1(PTX) vs. NC |
| Paclitaxel (PTX) [†] | 12.75 | 12.37 | 20.22 | 1.63 | 16.03 | 15.20 | 45.86 | **3.02** | 1.26 | 1.23 | **2.27** | **3.71** |
| Mitoxantrone (MXT) [†] | 4.31 | 4.11 | 6.79 | 1.65 | 4.23 | 4.34 | 11.21 | **2.58** | 0.98 | 1.06 | 1.65 | **2.73** |
| Doxorubicin (DXR) [†] | 4.03 | 4.34 | 6.67 | 1.54 | 4.24 | 4.15 | 10.43 | **2.51** | 1.05 | 0.96 | 1.56 | **2.40** |
| Vincristine (VCT) [†] | 6.72 | 7.15 | 12.61 | 1.76 | 6.96 | 7.29 | 21.51 | **2.95** | 1.04 | 1.02 | 1.71 | **3.01** |
| Ivermectin (IVM) [†] | 15.52 | 15.29 | 24.02 | 1.57 | 14.51 | 14.64 | 36.37 | **2.48** | 0.93 | 0.96 | 1.51 | **2.38** |
| Cyclosporin A (CSA) | 8.60 | 8.87 | 8.69 | 0.98 | 8.75 | 8.82 | 9.05 | 1.03 | 1.02 | 0.99 | 1.04 | 1.02 |
| Daunorubicin (DRC) | 3.91 | 4.16 | 3.95 | 0.95 | 3.88 | 4.05 | 10.23 | **2.53** | 0.99 | 0.97 | 2.59 | **2.46** |
| Loperamide (LPM) | 15.96 | 16.08 | 16.49 | 1.03 | 16.42 | 15.17 | 15.34 | 1.01 | 1.03 | 0.94 | 0.93 | 0.95 |
| Nitazoxanide (NTZ) [†] | 26.75 | 25.90 | 25.28 | 0.98 | 25.37 | 26.76 | 25.59 | 0.96 | 0.95 | 1.03 | 1.01 | 0.99 |

* Numbers in these columns are ratios (fold changes) of $TC_{50}$ values between specified cell lines on specified compounds. Bold fonts indicate those showing >2-fold increases of $TC_{50}$ values. [†] These compounds were selected for comparing their efficacies against the growth of *C. parvum* cultured in MDR1(PTX) and NC cell lines for determining on-target effects.

FDA-approved drug to treat human cryptosporidiosis, for which the mechanism of action still remained undefined. By applying continuous drug pressures of NTZ, all three resulting cell lines developed resistance to NTZ at varied levels (Table 3; Fig 3E and 3F). There were 1.45-, 1.51- and 2.06-fold increases of $TC_{50}$ values in WT(NTZ), NC(NTZ) and MDR1(NTZ) cells over their parental WT, NC and MDR1 cells (Table 3, right four columns). NTZ-selection did not increase the drug tolerance of WT, NC and MDR1 cells to the other eight compounds (Table 3), indicating that the developed resistance was specific for NTZ, rather than to multiple drugs.

Overexpression of *MDR1* and selection with PTX or NTZ caused no apparent changes on the morphology and growth of host cells in vitro (Fig 4). Selection with either PTX or NTZ had no significant effects on MDR1 protein levels as shown by immunofluorescence assay (IFA) (Fig 4). Western blot analysis also showed that the ratios of MDR1 protein levels for the three pairs of cell lines [i.e., MDR1 vs. NC; MDR1(PTX) vs. NC(PTX) and MDR1(NTZ) vs. NC(NTZ)] were relatively consistent (Fig 2E). Only the mRNA levels showed a relatively higher increase by the PTX-selection [i.e., MDR1(PTX) vs. NC(PTX)] (Fig 2F). How host cells after selection increased tolerance to PTX or NTZ with no significant increase of MDR1 protein level remains to be determined. One possible explanation currently under investigation is the mutations in endogenous and/or transgenic *MDR1* gene as demonstrated by other investigators [22].

In short summary, stable overexpression of *MDR1* in HTC-8 cells could increase tolerance of the cells to multiple MDR1 substrates (e.g., PTX), but at lower than 2-fold increase in general. The drug tolerance could be further increased by applying drug pressure. More importantly, stable overexpression of *MDR1* allowed the development of drug tolerance of host cells

**Table 3. Drug tolerance profiles of the three host cell lines before and after selection by nitazoxanide (NTZ) as determined by MTS assay and expressed in 50% inhibition concentrations ($TC_{50}$ values).**

| Compounds | $TC_{50}$ in cells without drug selection (µM) and ratios to NC | | | | $TC_{50}$ in cells with selection by NTZ (µM) and ratios to NC(NTZ) | | | | Fold increases of $TC_{50}$ after selection by NTZ (vs. their parent cell lines) | | | |
|---|---|---|---|---|---|---|---|---|---|---|---|---|
| | WT | NC | MDR1 | MDR1/ NC* | WT (NTZ) | NC (NTZ) | MDR1 (NTZ) | MDR1(NTZ)/ NC(NTZ)* | WT(NTZ) vs. WT | NC(NTZ) vs. NC | MDR1(NTZ) vs. MDR1 | MDR1(NTZ) vs. NC |
| Nitazoxanide (NTZ) [†] | 26.75 | 25.90 | 25.28 | 0.98 | 38.70 | 39.09 | 52.19 | 1.34 | 1.45 | 1.51 | **2.06** | **2.02** |
| Paclitaxel (PTX) | 12.75 | 12.37 | 20.22 | 1.63 | 12.51 | 12.73 | 21.94 | 1.72 | 0.98 | 1.03 | 1.09 | 1.77 |
| Mitoxantrone (MXT) | 4.31 | 4.11 | 6.79 | 1.65 | 4.15 | 4.19 | 6.74 | 1.61 | 0.96 | 1.02 | 0.99 | 1.64 |
| Doxorubicin (DXR) | 4.03 | 4.34 | 6.67 | 1.54 | 4.18 | 4.21 | 6.90 | 1.64 | 1.04 | 0.97 | 1.03 | 1.59 |
| Vincristine (VCT) | 6.72 | 7.15 | 12.61 | 1.76 | 7.04 | 6.87 | 12.38 | 1.80 | 1.05 | 0.96 | 0.98 | 1.73 |
| Ivermectin (IVM) | 15.52 | 15.29 | 24.02 | 1.57 | 14.75 | 15.18 | 22.59 | 1.49 | 0.95 | 0.99 | 0.94 | 1.48 |
| Cyclosporin A (CSA) | 8.60 | 8.87 | 8.69 | 0.98 | 9.12 | 9.06 | 8.71 | 0.96 | 1.06 | 1.02 | 1.00 | 0.98 |
| Daunorubicin (DRC) | 3.91 | 4.16 | 3.95 | 0.95 | 4.09 | 3.85 | 3.93 | 1.02 | 1.05 | 0.93 | 0.99 | 0.94 |
| Loperamide (LPM) | 15.96 | 16.08 | 16.49 | 1.03 | 16.23 | 15.64 | 15.35 | 0.98 | 1.02 | 0.97 | 0.93 | 0.95 |

* Numbers in these columns are ratios (fold changes) of $TC_{50}$ values between specified cell lines on specified compounds. Bold fonts indicate those showing >2-fold increases of $TC_{50}$ values. [†] The compound was used in subsequent experiments to assess their efficacies against the growth of *C. parvum* cultured in MDR1(NTZ) and NC cell lines for determining on/off-target effects.

to non-MDR1 substrates (e.g., NTZ) by applying drug pressure in a relatively short timeframe (e.g., in around three months to develop >2.0-fold increase of resistance to NTZ).

## PTX and NTZ inhibited the growth of *C. parvum* by acting fully on the parasite targets, while DXR, IVM, MXT and VCT acted on both the parasite and host targets

The availability of host cells with >2 to 3-fold increase of drug resistance to NTZ, PTX and four other compounds made it possible to evaluate whether, and how much, the anti-cryptosporidial activities of these compounds were attributed to their actions on the parasite targets. In theory, if a specified inhibitor inhibited the epicellular *C. parvum* in vitro by solely acting on the parasite target and its action on host cell target made no contribution to the antiparasitic activity, the increase of resistance to the inhibitor in the host cells would not affect the anti-cryptosporidial activity [15]. This could be achieved by comparing the anti-cryptosporidial efficacy ($EC_{50}$ values) with cytotoxicity ($TC_{50}$ values) of the inhibitor between MDR1 (PTX) and NC or between MDR1(NTZ) and NC cells.

As a quality control, we first confirmed that overexpression of *MDR1* and drug selection by either PTX or NTZ in the host cells had no effect on the infection and growth of *C. parvum* by a qRT-PCR-based 44-h infection assay, in which all nine cell lines showed virtually identical parasite loads (Fig 5A). There were no enrichment of MDR1 protein at the *C. parvum* infection sites in MDR1, MDR1(PTX) and MDR1(NTZ) cells (Fig 5B), indicating the overexpression of MDR1 would not alter the drug fluxes at the host cell-parasite interface or on the parasitophorous vacuole membrane (PVM) to complicate the drug action on the parasite. All compounds displayed no difference in their anti-cryptosporidial activities between WT and NC cells, confirming that transfection of cells with vector alone had no effect on the action of compounds

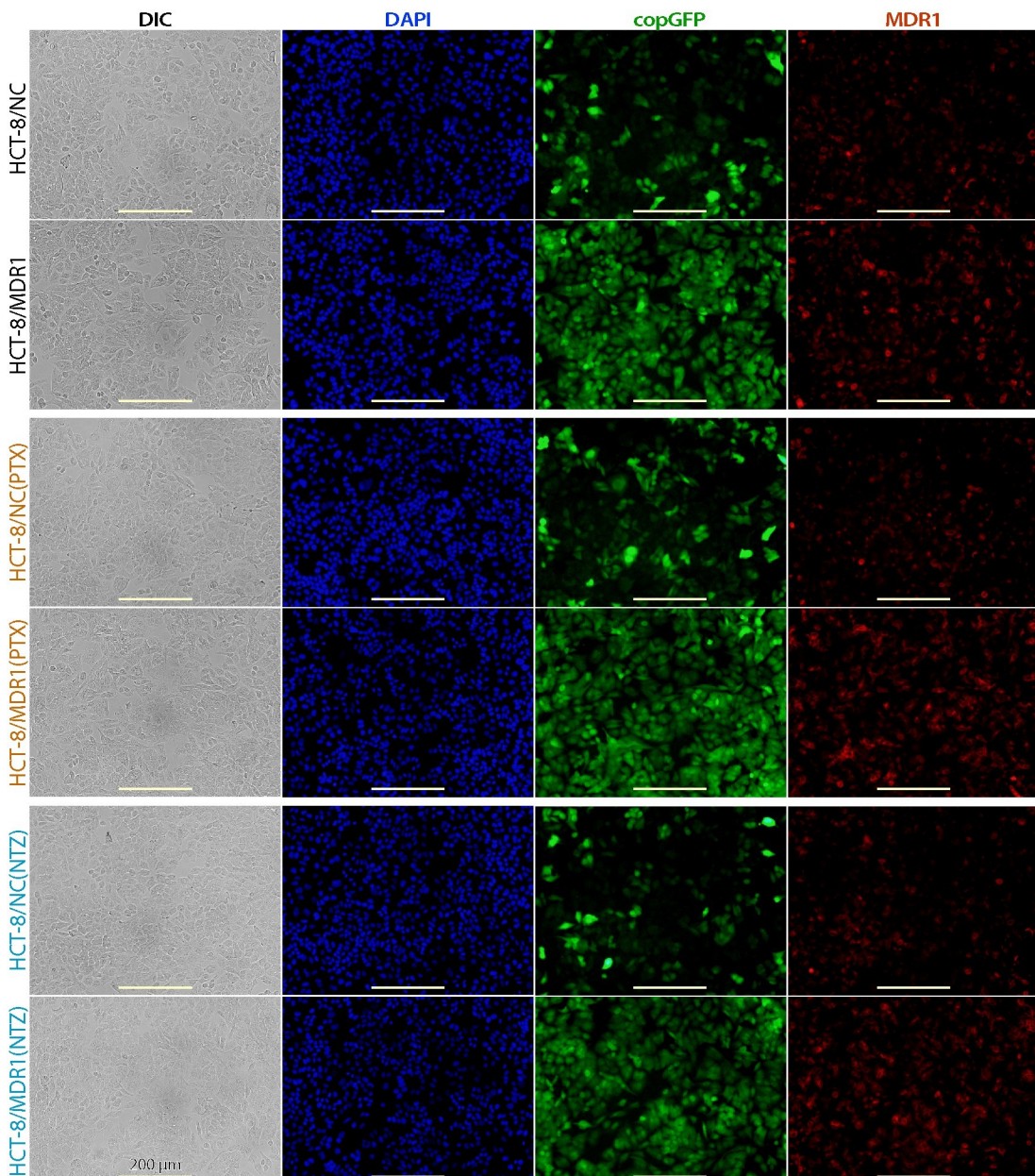

**Fig 4. Morphology of HCT-8/NC, HCT-8/MDR1, HCT-8/NC(PTX), HCT-8/MDR1(PTX), HCT-8/NC(NTZ), HCT-8/MDR1 (NTZ) cells.** Cells were cultured for 24 h until near confluence, showing that overexpression of *MDR1* and drug selections by paclitaxel (PTX) or nitazoxanide (NTZ) had no apparent effect on the morphology and growth of host cells. See Table 1 for more detailed descriptions about the cell lines. DIC, differential interference contrast microscopy; DAPI, 4′,6-diamidino-2-phenylindole for counterstaining nuclei; copGFP, copepod green fluorescence protein present in both blank control and *MDR1*-carrying vector; MDR1, multidrug resistance protein-1 protein labeled by immunostaining.

to the parasite (Fig 6; Table 4). We then used these host cell lines to evaluate the on/off-target effects of PTX, NTZ and four other compounds by examining whether increased drug tolerance in host cells affected anti-cryptosporidial activities.

Using MDR1(PTX) cells as an in vitro model (vs. WT and NC cells), PTX showed the same anti-cryptosporidial efficacy in all three cell lines based on the inhibitory curves and $EC_{50}$ values (Fig 6A; Tables 4 and S3), indicating that the increase of drug tolerance in host cells had no

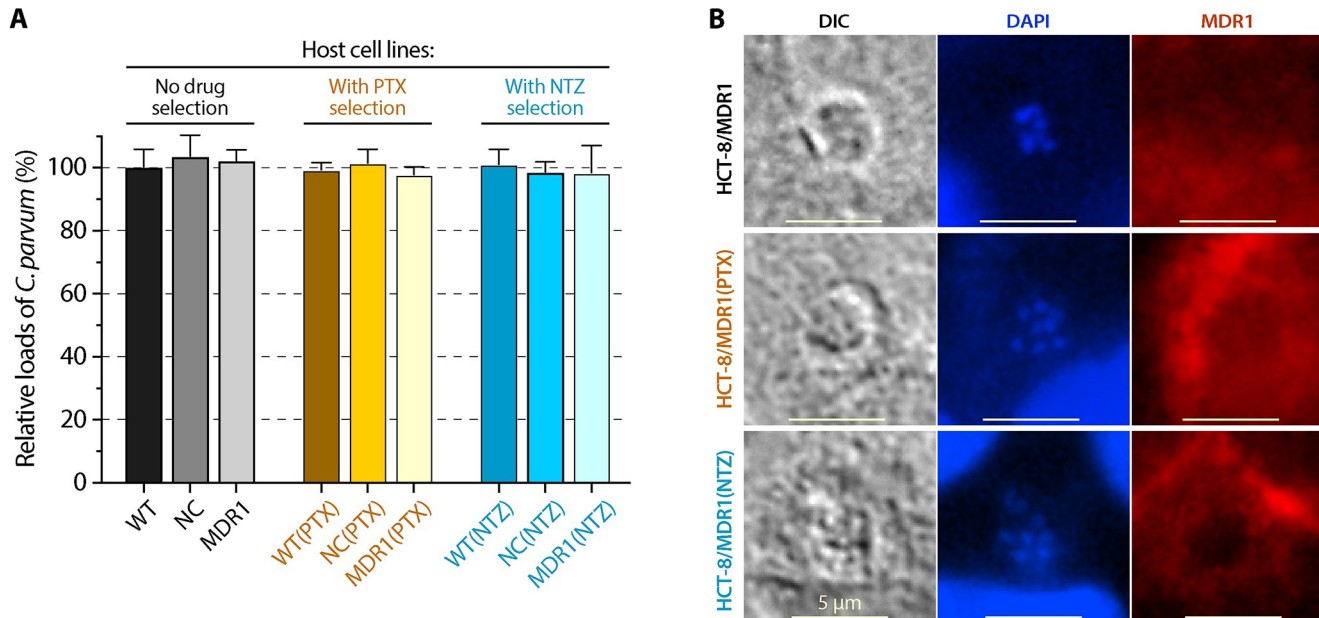

**Fig 5. Effect of MDR1-overexpression and drug selection with paclitaxel (PTX) and nitazoxanide (NTZ) in host cells on the infection of *Cryptosporidium parvum* in vitro.** (A) Relative loads of *C. parvum* grown on the nine host cell lines as determined by 44-h infection assay followed by qRT-PCR detection of the parasite 18S rRNA (Cp18S). The relative levels of Cp18S were normalized with host cell 18S rRNA (Hs18S) and expressed as the percent levels using that from WT cells as the baseline. The data showed that overexpression of MDR1 and selection by PTX or NTZ had no or little effect on the parasite infection. Bars represent the standard errors of the means (SEMs; $n = 3$). (B) Immunostaining of MDR1 in the three MDR1-overexpressing host cell lines that were infected with *C. parvum* for 24 h, showing no particular accumulation of MDR1 protein at the infection sites.

effect on the killing of the parasite by PTX. This confirmed that PTX inhibited the parasite growth by solely acting on the parasite target (i.e., 100% on-target). The data agreed with previous observation using transient overexpression models [15]. For the other four compounds to which MDR1(PTX) cells also developed >2-fold increase of drug tolerance (i.e., DXR, IVM, MXT and VCT), their $EC_{50}$ values increased by 31.6% to 103.9% (vs. NC cells) (Fig 6B–6E; Table 4), meaning that the increase of drug tolerance affected the anti-cryptosporidial efficacies of the four compounds and their actions on host cells (i.e., off-target effect) also contributed to the killing of the parasite. Because the percent increases of $EC_{50}$ values were less than those of $TC_{50}$ values (e.g., for MXT, the percent change of $EC_{50}$ was 103.9% while that of $TC_{50}$ was 172.7%) (Table 4), we might conclude that the off-target effects contributed partially to the killing of the parasite by the four inhibitors. In other words, both on-target and off-target effects contributed to the anti-cryptosporidial activities of the four inhibitors.

The development of NTZ-resistant cell line [i.e., MDR1(NTZ) cells] allowed us to evaluate the on-target effect of NTZ against *C. parvum* for the first time since its anti-cryptosporidial activity was discovered. In this assay, the increase of drug tolerance in host cells had no effect on the antiparasitic activity of NTZ based on the inhibitory curves and $EC_{50}$ values between WT, NC and MDR1(NTZ) cells (Fig 6F; Tables 4 and S3). This result confirmed that, like PTX, NTZ inhibited the *C. parvum* growth by fully acting on the parasite target (i.e., 100% on-target).

### Full or partial on-parasite-target effects were further validated using the MDR1 inhibitor elacridar

If the increase of tolerance to a specified inhibitor in host cells was truly mediated by MDR1, specific inhibition of MDR1 would restore the sensitivity of the host cells to the inhibitor (as

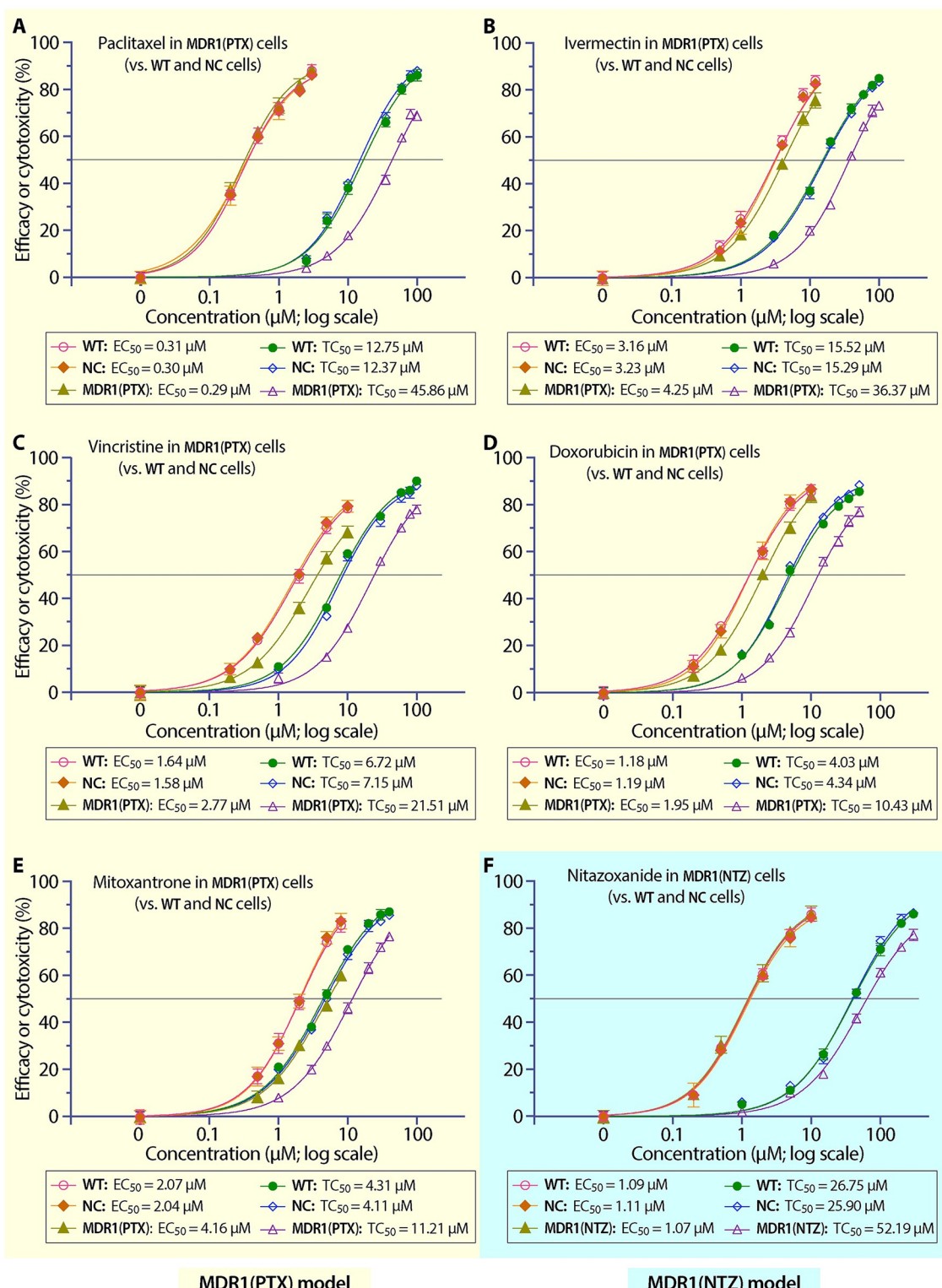

**Fig 6. Evaluation of the on/off-target effects of the six inhibitors under investigation based on anti-cryptosporidial efficacy and cytotoxicity curves and 50% inhibition values.** (A–E) Efficacy and cytotoxicity of paclitaxel (A), ivermectin (B), vincristine (C), doxorubicin (D) and mitoxantrone (E) using MDR1(PTX) cell model (vs. WT and NC cells). (F) Efficacy and cytotoxicity of nitazoxanide using MDR1(NTZ) cell model (vs. WT and NC cells). In all panels, there were no or little differences on the efficacy or cytotoxicity curves between WT and NC cells. Increased drug tolerance (i.e., reduced cytotoxicity) to paclitaxel (A) or nitazoxanide

(F) in host cells had no effect on the anti-cryptosporidial efficacy, while increased drug tolerance (reduced cytotoxicity) to ivermectin (B), vincristine (C), doxorubicin (D) and mitoxantrone (E) in host cells reduced the anti-cryptosporidial efficacy. Bars represent the standard errors of the means (SEMs; $n = 3$).

indicated by $TC_{50}$). Additionally, inhibition of MDR1 would not affect the anti-cryptosporidial efficacy ($EC_{50}$) if the drug that acted only on the parasite target, or partially affect the efficacy ($EC_{50}$) if the drug that also acted on host cell target. This notion was tested using elacridar, a third generation of MDR1 inhibitor [18,23–25]. The concentration of elacridar at 300 nM was used based on previous studies that elacridar at this concentration could produce strong inhibition on the activity of MDR1 with no significant cytotoxicity to HCT-8 cells [15]. This study also confirmed that elacridar at 300 nM produced no or little effect on the growth of the six cell lines and on the growth of *C. parvum* cultured with NC, MDR1(PTX) and MDR1(NTZ) cells (Fig 7). We then examined the effect of elacridar on the cytotoxicity and anti-cryptosporidial efficacies of the six inhibitors in the range of $TC_{50}$ or $EC_{50}$ concentrations.

In NC, MDR1, NC(PTX) and MDR1(PTX) cells, elacridar dramatically reduced the tolerance of these cells to PTX, MXT, DXR, VCT and IVM (Fig 8A and 8B), indicating that: 1) the increased tolerance to the five inhibitors in MDR1 and MDR1(PTX) cells was MDR1-dependent; and 2) there was a basal level of MDR1 in NC and NC(PTX) cells (negative controls) that could be inhibited by elacridar (Fig 8A and 8B, columns in black). However, elacridar had no effect on the anti-cryptosporidial activity of PTX in both NC and MDR1(PTX) cells (Fig 8C), confirming that the killing of *C. parvum* by PTX was unrelated to the MDR1 activity. In other words, the action of PTX on the host cell target had no effect on anti-cryptosporidial activity by PTX, so that the killing of *C. parvum* was solely attributed to its action on the parasite target (100% on-target). On the other hand, elacridar reduced the anti-cryptosporidial activities of MXT, DXR, VCT and IVM in both NC and MDR1(PTX) cells, indicating that the killing of *C. parvum* by the four inhibitors was associated with MDR1 activity, so the actions of the four inhibitors on the host cell targets also contributed to the inhibition of the growth of *C. parvum*.

**Table 4. Anti-cryptosporidial efficacies ($EC_{50}$) of selected compounds in specified cell lines in comparison with corresponding drug resistance parameters and on/off-target rates calculated based on the ratios of percent (Pct) changes between $EC_{50}$ and $TC_{50}$ values\*.**

| Data obtained from MDR1(PTX) cell-based host cell model | | | | | | | | | | |
|---|---|---|---|---|---|---|---|---|---|---|
| **Compound** | **Antiparasitic efficacy ($EC_{50}$) (µM)** | | | | **Host cell cytotoxicity ($TC_{50}$) (µM)** | | | | **On/off-target rate at $EC_{50}$** | |
| | **WT** | **NC** | **MDR1(PTX)** | **Pct change (vs NC)** | **WT** | **NC** | **MDR1(PTX)** | **Pct change (vs NC)** | **Off-target ($E_{off}$)** | **On-target ($E_{on}$)** |
| Paclitaxel (PTX) | 0.31 | 0.30 | 0.29 | -3.3% | 12.75 | 12.37 | 45.86 | 270.7% | -1.2% | 101.2% |
| Mitoxantrone (MXT) | 2.07 | 2.04 | 4.16 | 103.9% | 4.31 | 4.11 | 11.21 | 172.7% | 60.2% | 39.8% |
| Doxorubicin (DXR) | 1.18 | 1.19 | 1.95 | 63.9% | 4.03 | 4.34 | 10.43 | 140.3% | 45.5% | 54.5% |
| Vincristine (VCT) | 1.64 | 1.58 | 2.77 | 75.3% | 6.72 | 7.15 | 21.51 | 200.8% | 37.5% | 62.5% |
| Ivermectin (IVM) | 3.16 | 3.23 | 4.25 | 31.6% | 15.52 | 15.29 | 36.37 | 137.9% | 22.9% | 77.1% |
| Data obtained from MDR1(NTZ) cell-based host cell model | | | | | | | | | | |
| **Compound** | **WT** | **NC** | **MDR1(NTZ)** | **Pct change (vs NC)** | **WT** | **NC** | **MDR1(NTZ)** | **Pct change (vs NC)** | **Off-target rate (%)** | **On-target rate (%)** |
| Nitazoxanide (NTZ) | 1.09 | 1.11 | 1.07 | -3.6% | 26.75 | 25.90 | 52.19 | 101.5% | -3.6% | 103.6% |

\*Percent (Pct) changes refer to the changes of $EC_{50}$ or $TC_{50}$ values in MDR1(X) cells (X = PTX or NTZ) in comparison to NC cells calculated using following formulae: *Pct change of* $EC_{50} = (EC_{50(MDR1(X))} - EC_{50(NC)})/EC_{50(NC)}$; *Pct change of* $TC_{50} = (TC_{50(MDR1(X))} - TC_{50(NC)})/TC_{50(NC)}$. The $E_{on}$ and $E_{off}$ refer the on-target and off-target effect calculated using formula: $E_{off} = (Pct\ change\ of\ EC_{50})/(Pct\ change\ in\ TC_{50}) \times 100\%$; $E_{on} = (1 - E_{off}) \times 100\%$. Also see S3 Table for a list of detailed parameters and values, including selectivity index (SI) and the ratios of SI between drug-selected and negative control cell lines.

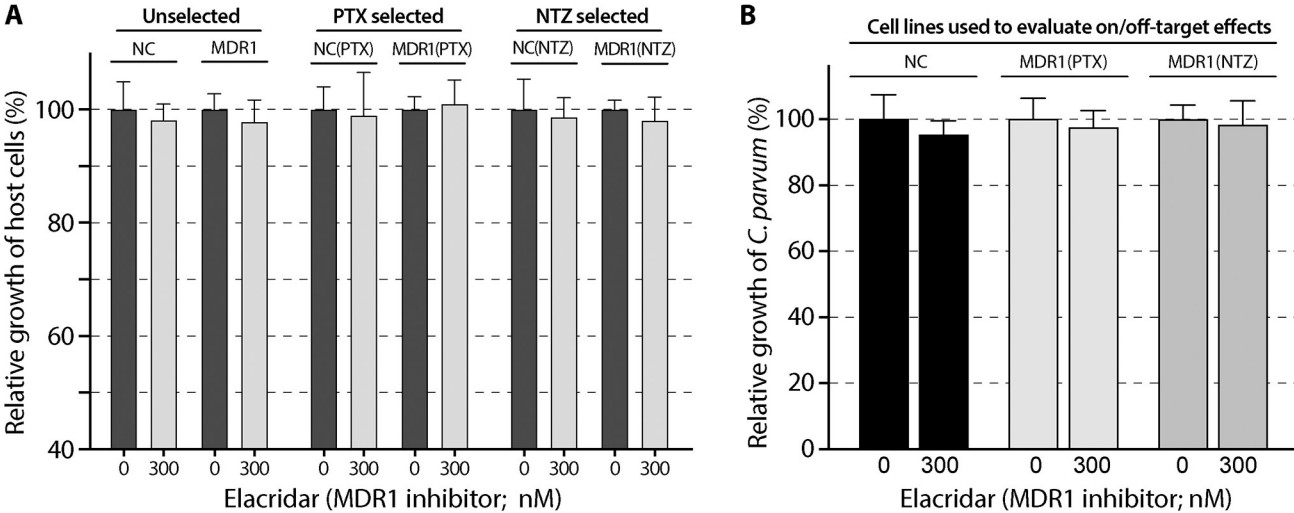

**Fig 7. Effects of the MDR1 inhibitor elacridar (300 nM) on the growth of the six host cell lines including NC, MDR1, NC(PTX), MDR1(PTX), NC (NTZ) and MDR1(NTZ) cells (A) and on the growth of *Cryptosporidium parvum* cultured with NC, MDR1(PTX) and MDR1(NTZ) cells (B).** The host cell growth was determined by MTS cytotoxicity assay. The parasite growth was determined by 44-h infection/qRT-PCR assay. See Table 1 for more detailed descriptions about the cell lines. Bars represent the standard errors of the means (SEMs; *n* = 3). There was no statistical significances between each pair of specimens (i.e., 300 nM vs. 0 nM elacridar) by Holm-Šídák multiple *t*-test.

In the case of NTZ, elacridar had little effect on the cytotoxicity of NTZ in NC(NTZ) cells (Fig 9A). This observation agreed with the fact that NTZ was not an MDR1 substrate, so that its cytotoxicity should not be affected by the inhibition of basal level MDR1. The tolerance to NTZ in MDR1(NTZ) cells was reverted by elacridar (Fig 9A), indicating that the NTZ-resistance developed in MDR1(NTZ) cells was related to overexpressed MDR1. In the efficacy assay, elacridar had no effect on the anti-cryptosporidial activity of NTZ in both NC and MDR1(NTZ) cells (Fig 9B), confirming the killing of *C. parvum* by NTZ was unrelated to the MDR1 activity, or in other words, the action of NTZ on the host cell target made no contribution to the killing of *C. parvum*.

## Quantitative estimation of the relative contributions of on-target and off-target effects to the observed anti-cryptosporidial activity

**Estimation of on-target rate based on $EC_{50}$ and $TC_{50}$ values.** We showed that stable *MDR1*-transgenic cells could increase drug tolerance to MDR1 substrates or non-substrates in response to selection. These cell lines could serve as an in vitro model to assess whether an anti-cryptosporidial compounds killed *C. parvum* via acting fully (PTX and NTZ) or partially (MXT, DXR, VCT and IVM) on the parasite targets. We were also intrigued in quantifying the proportions of contributions of on-target effect (i.e., the action on the parasite target) to the antiparasitic activity. Because $EC_{50}$ and $TC_{50}$ were the most commonly used parameters for drug efficacy and cytotoxicity, we first attempted to develop a formula to calculate the on-target effect based on $EC_{50}$ and $TC_{50}$ values. The theory was that, the effect on the host target (i.e., off the parasite target) at 50% efficacy (denoted as $E_{50(off)}$) was correlated to the ratio between the relative increase (or percent increase) of $EC_{50}$ and the relative increase of $TC_{50}$, which could be calculated using the equation (see the equation derivations in the Methods section):

$$E_{50(\text{off})} = \left(\frac{EC_{50(\text{MDR1})} - EC_{50(\text{NC})}}{EC_{50(\text{NC})}}\right) / \left(\frac{TC_{50(\text{MDR1})} - TC_{50(\text{NC})}}{TC_{50(\text{NC})}}\right) \times (100\%) \qquad (1)$$

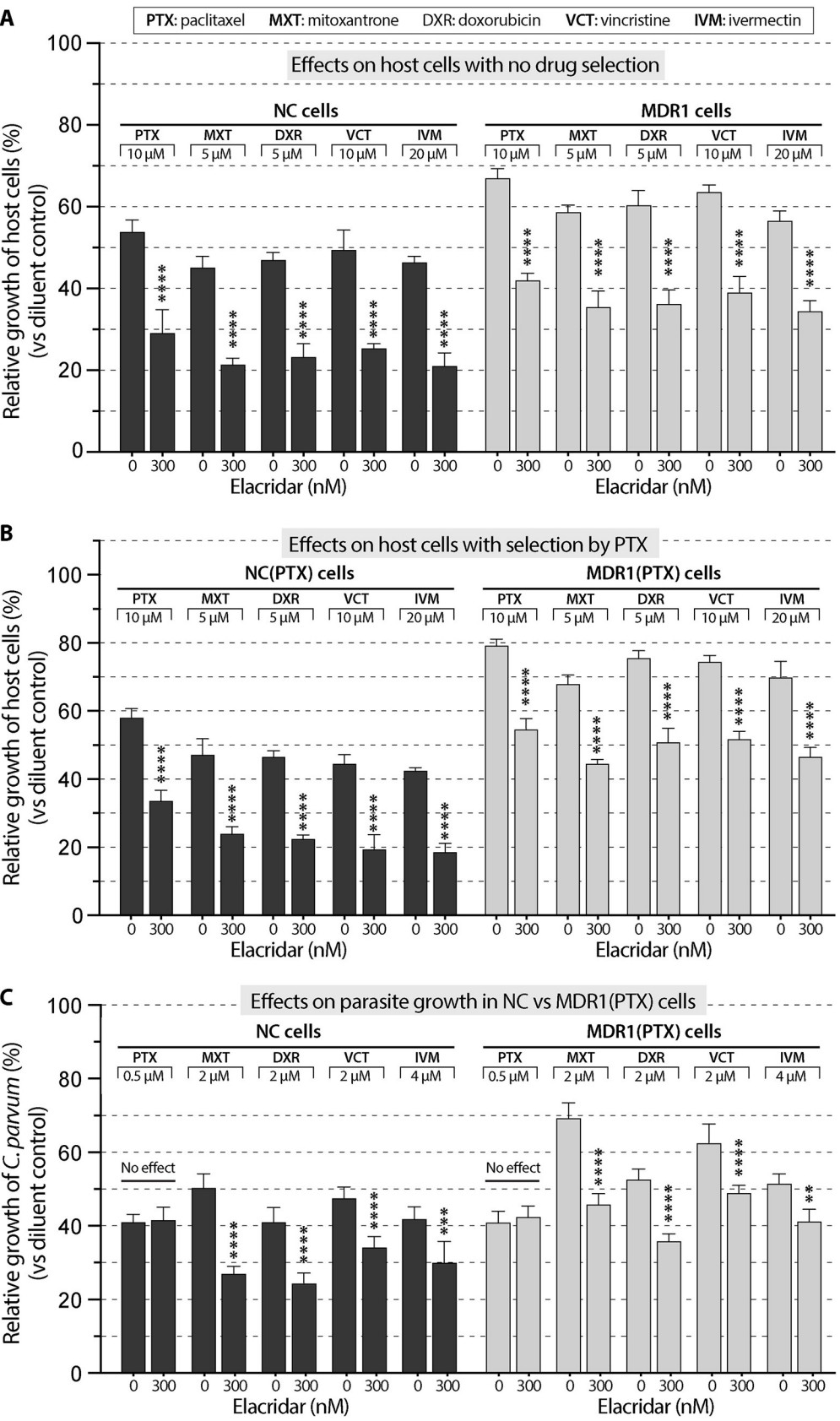

**Fig 8. Effect of MDR1-inhibition by elacridar (300 nM) on the cytotoxicity and anti-cryptosporidial activity of the five inhibitors using MDR1(PTX) cell model.** (A) Effect of elacridar on the cytotoxicity of the five inhibitors on NC and MDR1 cells. (B) Effect of elacridar on the cytotoxicity of the five inhibitors on NC(PTX) and MDR1(PTX) cells. (C) Effect of elacridar on the anti-cryptosporidial activity of the five inhibitors against *Cryptosporidium parvum* cultured on NC and MDR1(PTX) cells. Cytotoxicity of inhibitors at specified concentrations on the host cells was evaluated by MTS cytotoxicity assay. Anti-cryptosporidial activity of inhibitors at specified concentrations was determined by 44-h infection/qRT-PCR assay. PTX, paclitaxel; MXT, mitoxantrone; DXR, doxorubicin; VCT, vincristine; IVM, ivermectin. Bars represent the standard errors of the means (SEMs; *n* = 3). Statistical significances were evaluated by Holm-Šídák multiple *t*-test between group pairs (** = *p* <0.01, *** = *p* <0.001 and **** = *p* <0.0001).

while the on-target rate at 50% efficacy (denoted as $E_{50(on)}$) could be calculated by:

$$E_{50(on)} = (1 - E_{50(off)}) \times (100\%) \qquad (2)$$

Using Eqs 1 and 2, we obtained theoretical on/off-target rates for the six compounds, in which the on-target rates for PTX and NTZ were 101.2% and 103.6%, respectively (Table 4). The values were slightly higher than 100% (the theoretical maximum) due to the assaying errors. The other four compounds varied in their on/off-target rates, i.e., on-target rate 39.8% for MXT, 54.5% for DXR, 62.5% for VCT and 77.1% for IVM (Table 4). It was noticeable that the off-target effects of MXT and DXR contributed more than 50% or near 50% to the observed anti-cryptosporidial activity at 50% efficacy.

**Estimation of on/off-target rates in the whole efficacy range.** In theory, an inhibitor at different concentrations might act at varied levels on the parasite target and host cell target. In other words, an inhibitor's on/off-target effects might differ in their contributions to the anti-parasitic activity at varied efficacy levels. We denote $E_{i(on)}$ or $E_{i(off)}$ as the on-target or off-target rate for a compound at $EC_i$ (the concentration of the compound inhibiting the parasite growth by *i*%; *i* = 0 to 100). The off-target rate $E_{i(off)}$ at the specified efficacy $EC_i$ could be estimated

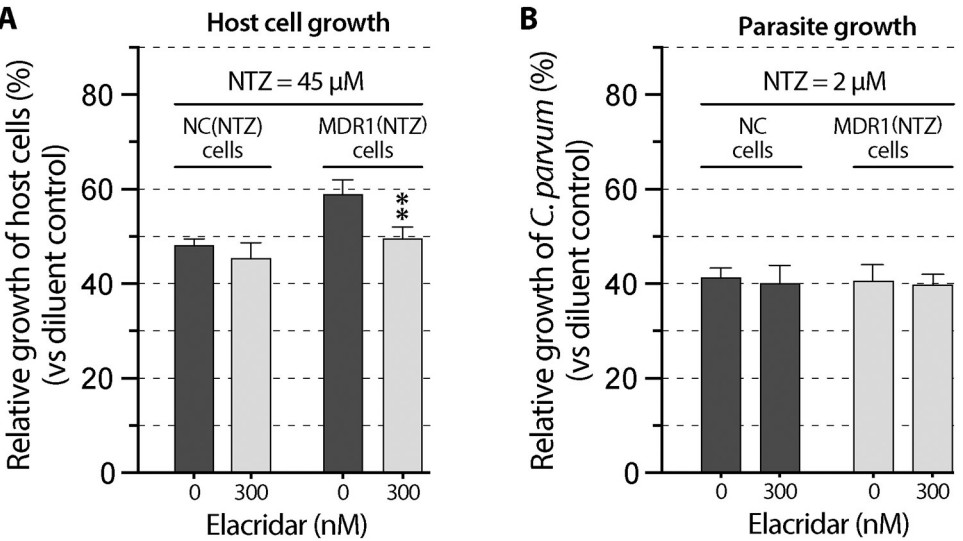

**Fig 9. Effect of MDR1-inhibition by elacridar (300 nM) on the cytotoxicity and anti-cryptosporidial activity of nitazoxanide (NTZ) using MDR1(NTZ) cell model.** (A) Effect of elacridar on the cytotoxicity of NTZ on NC(NTZ) and MDR1(NTZ) cells. (B) Effect of elacridar on the anti-cryptosporidial activity of NTZ against *Cryptosporidium parvum* cultured on NC and MDR1(NTZ) cells. Bars represent the standard errors of the means (SEMs; *n* = 3). Statistical significances were evaluated by Holm-Šídák multiple *t*-test between group pairs (** = *p* <0.01).

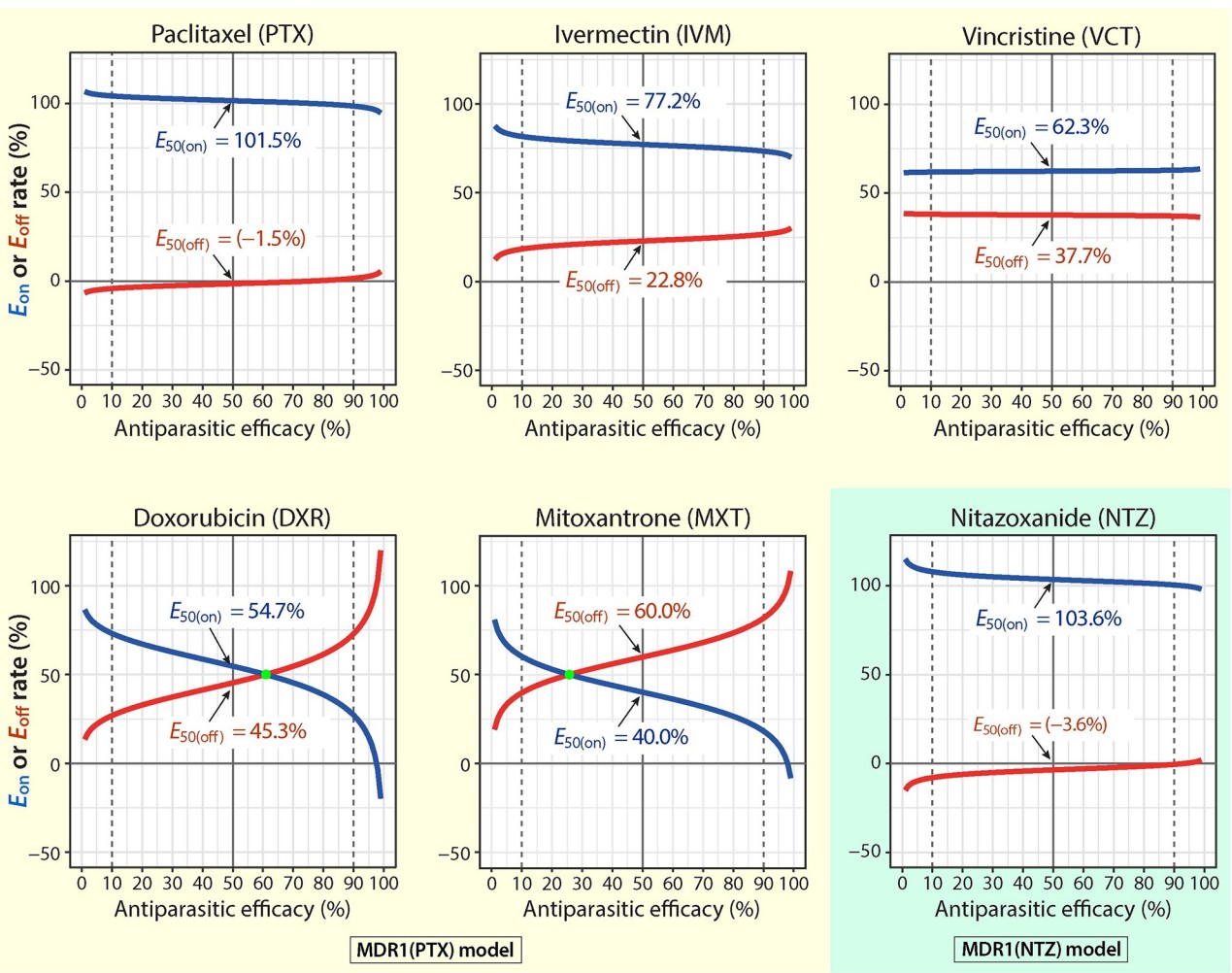

**Fig 10. Percent contributions (rates) of on-target and off-target effects to the observed anti-cryptosporidial activity of paclitaxel (PTX), ivermectin (IVM), vincristine (VCT), doxorubicin (DXR), mitoxantrone (MXT) and nitazoxanide (NTZ) across the range of anti-cryptosporidial efficacy.** The on-target ($E_{on}$) or off-target ($E_{off}$) rate refers to the percent contribution of the action of a specified inhibitor on the parasite target or on the host cell target to the observed anti-cryptosporidial activity as calculated using Eq 3 and 4. Parameters (i.e., $EC_{50}$, $TC_{50}$ and Hill slope $h$) were obtained by nonlinear regressions of the same datasets for plotting the cytotoxicity and efficacy curves in Fig 6. The $E_{50(on)}$ and $E_{50(off)}$ values shown in each plot refer to the on-target and off-target rates of a specified inhibitor at the $EC_{50}$ concentration.

using the following Eq 3 that was generalized from Eq 1:

$$E_{i(off)} = \left( \frac{EC_{i(MDR1)} - EC_{i(NC)}}{EC_{i(NC)}} \right) / \left( \frac{TC_{i(MDR1)} - TC_{i(NC)}}{TC_{i(NC)}} \right) \times (100\%) \qquad (3)$$

while the on-target rate could be calculated by:

$$E_{i(on)} = (1 - E_{i(off)}) \times (100\%) \qquad (4)$$

where $EC_i$ and $TC_i$ ($i = 0$ to $100$) values were calculated using a 4-parameter logistic (4PL) model (see Eq 9 and derivation in the Methods). Using Eq 3 and 4, we were able to plot the on/off-target rates for the six inhibitors over the entire efficacy range (Fig 10). Based on the relative linearity of the 4PL model-derived sigmoidal curves (Fig 10), the on/off-target rates between $EC_{10}/TC_{10}$ and $EC_{90}/TC_{90}$ might be considered more reliable and biologically relevant

(also see S5 Table for representative $E_{on}$ and $E_{off}$ values between $EC_{10}$ and $EC_{90}$). An open source Python code to plot the $E_{on(i)}$ and $E_{off(i)}$ curves from parameters $EC_{50}$, $TC_{50}$ and Hill slope $h$ was developed and deposited at the GitHub depository (https://github.com/alienn233/PACOOTER).

In the plots, both PTX and NTZ produced relatively parallel curves of $E_{on}$ and $E_{off}$ near 100% and 0%, respectively (Fig 10A and 10F). The $E_{on}$ and $E_{off}$ curves for IVM and VCT were also relatively parallel, showing higher contributions from the on-target effects ($E_{10(on)}$ to $E_{90(on)}$ values = 81.6% to 73.4% for IVM, and 61.9% to 62.9% for VCT) (Fig 10B and 10C; S5 Table). There was a small surprise for DXR and MXT that showed lowest the $E_{50(on)}$ values as described above, in which the $E_{on}$ and $E_{off}$ curves were non-parallel and intersected (Fig 10D and 10E). The curves for DXR and MXT revealed that, at upper or lower effective concentrations, the on-target effect contributed more to the antiparasitic activity of DXR and MXT, but this trend was reversed at higher effective concentrations after concentrations reached to certain points (i.e., at $E_{62}$ and $E_{27}$, respectively).

## Relationships between *in vitro* selectivity index (*SI*), on-target ratio and cytotoxicity

It was noticed that an on-target inhibitor would have a larger selectivity index (*SI*; or $SI_{50}$ for accuracy as it was determined by the $TC_{50}/EC_{50}$ ratio) [15]. Here we further observed a certain linear relationship between $E_{50(on)}$ and *SI* in WT cells for the four partially on-target inhibitors (Fig 11A, green line). The *SI* values of the four inhibitors were all in single digits (i.e., *SI* = 0.21, 3.42, 4.1 and 4.91 for MXT, DXR, VCT and IVM, respectively) in comparison to those in double digits for NTZ and PTX (i.e., *SI* = 24.54 and 41.13, respectively). When nonlinear regressions were applied to all six compounds, the relationship between $E_{50(on)}$ and *SI* roughly followed the 4PL model (Fig 11A, red line; $h$ = 1.891; $R^2$ = 0.9882). The authenticity of the nonlinear relationship remained to be confirmed after more values were available, particularly those in the upper quartile of $E_{50(on)}$ values. Apparently, the *SI* values for the four partially on-target and the two fully on-target inhibitors were separated by the "10-fold selectivity window" that was commonly used as criterion at the hit stage of drug discovery [26].

We further examined the mathematical relationship between *SI* and cytotoxicity, aiming to explore whether *SI* in WT cells might serve as a hint for the quality of hits. The assumption was that, for a fully on-target inhibitor, the cytotoxicity would be null or minimal in the range of concentrations showing antiparasitic efficacy. Based the 4PL model, the following equation was derived (see the derivation of equations in the Methods section):

$$T_{(Ei)} = \left( \frac{k^h}{E_i} - k^h + 1 \right)^{-1} \tag{5}$$

where $E_i$ denoted the antiparasitic efficacy [$i$ = 0 to 100(%)], $T_{(Ei)}$ denoted the cytotoxicity of the inhibitor at the concentration producing the efficacy $E_i$, $k$ represented selectivity index (SI) and $h$ represented the Hill slope (note that the $i$ values could only approach 0 or 100(%), but would never be equal to 0 or 100(%)). In this equation, $h$ values in the efficacy and cytotoxicity curves were set to be the same based on the assumption that a specified inhibitor would possess the same or similar mode of action on the parasite and host cells. The notion was supported in part by the actual $h$ values obtained in this study (S6 Table).

For the six compounds under investigation, both $k$ and $h$ were defined constants, thus allowing us to plot their relationship curves between the rates of calculated cytotoxicity and antiparasitic efficacy (Fig 11B). As expected, the cytotoxicity of all inhibitors rose along with the increase of efficacious concentrations but the trends were nonlinear and displayed as five

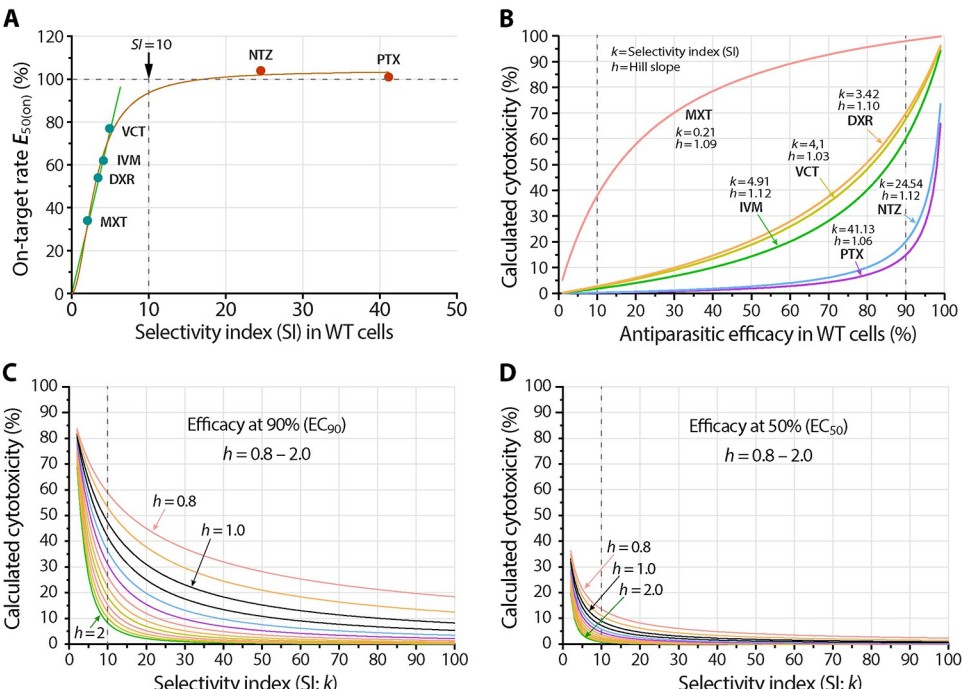

**Fig 11. Relationships between selectivity index (*SI* or *k*), on-target rate (*E*$_{50(\text{on})}$) and Hill slope (*h*).** (A) Plot of the on-target rate at *EC*$_{50}$ (*E*$_{50(\text{on})}$) against the selectivity index (*SI*) of the six inhibitors under investigation in wild-type (WT) cells. (B) Plot of the cytotoxicity for the six inhibitors under investigation (as percent inhibition on the growth of WT host cells) against the anti-cryptosporidial efficacy (as percent inhibition on the growth of *Cryptosporidium parvum* cultured with WT cells). The data showed the effect of parameters *k* and *h* on the curve shapes. (C, D) Plots of theoretical cytotoxicity in percent values and selectivity index (*SI* or *k*; values between 1 to 100) for curves with varied Hill slope values (*h* = 0.8 to 2.0) at two representative antiparasitic efficacy values, i.e., at *EC*$_{90}$ (C) and *EC*$_{50}$ (D). The data showed the effect of parameter *h* on the curve shapes.

concave and one convex rising curves. Curves were more skewed towards the two ends, e.g., at the 0–10% and 90–100% efficacy regions. Overall, the increase rates of the curves were negatively correlated to the SI and on-target rates, i.e., inhibitors with higher *k* and *E*$_{50(\text{on})}$ had a slower rate of increase of cytotoxicity, or vice versa. Apparently, the cytotoxicity values at *EC*$_{50}$ for the two on-target inhibitors NTZ and PTX were less than 5% (calculated values = 2.73% and 1.92%, respectively), while those for the four partially on-target inhibitors were much higher (i.e., between 14.40% and 20.56% for IVM, VCT and DXR, and up to 84.61% for MXT) (Fig 11B). Only MXT produced a convex curve due to the fact that its *SI* value was less than 1 (i.e., *k* = 0.21). In fact, *SI* was the determinant for the curve curvature (i.e., *k* = <1, 1, or >1 would produce convex, linear and concave curves, respectively).

Eq 5 also provided an opportunity to examine the relationship between an inhibitor's theoretical cytotoxicity and *k* (*SI*) and Hill slope (*h*) at specified efficacy (*E*$_i$). Based on the *h* values in this study (i.e., between 1.0 to 1.24), we plotted two sets of curves as examples to show the relationships between cytotoxicity, *k* (between 1 and 100) and *h* (between 0.8 and 2.0) at 50% and 90% efficacy concentrations (i.e., *EC*$_{50}$ and *EC*$_{90}$; the two commonly used parameters for drug efficacy) (Fig 11C and 11D). From the two plots, we observed that: 1) with fixed *h* and *k* values, an inhibitor's cytotoxicity was higher at higher efficacious concentrations (e.g., *T*$_{(E90)}$ > *T*$_{(E50)}$); 2) with fixed *h* and efficacy, all curves declined more sharply at lower *k* values, more apparently at *k* <10; and 3) with fixed *k* and efficacy, the cytotoxicity was negatively correlated with the *h* values (i.e., a lower toxicity at higher *h* value). Nonetheless, the plots gave us a new perspective to examine and compare the properties of inhibitors that might not be easily seen

from the efficacy/cytotoxicity curves and commonly used parameters (e.g., $EC_{50}$, $TC_{50}$ and $SI$ values). The effect of Hill slope ($h$) on cytotoxicity was a novel observation, although it might be noticeable by careful comparison between the cytotoxicity and efficacy curves (Fig 6).

## Discussion

The multidrug resistance protein 1 (MDR1; aka P-gp or ABCB1) is an ATP-dependent efflux pump with broad substrate specificity [16,17]. Both transient and stable overexpression of MDR1 in HCT-8 cells could increase the tolerance of cells to multiple compounds, but the two models have their own advantages and disadvantages. In the transient transfection model, host cells overexpressing MDR1 could be generated instantly to show >2-fold increases of drug tolerance to multiple compounds, but applicable only to MDR1 substrates [15]. In stable transfection model as demonstrated in this study, it might take months to first generate *MDR1*-transgenic cells that only showed <2-fold increases of drug tolerance. However, drug tolerance in *MDR1*-transgenic cells could be quickly increased to much higher than 2-fold by applying continuous drug pressure. The notable advantage of the stable transfection model is the potential to generate novel resistance to non-MDR1 substrates in a reasonably short timeframe that is otherwise unachievable using transient transfection model. As exemplified in this study, NTZ-resistance was generated in *MDR1*-transgenic HCT-8 cells in three months (>2-fold in MDR1(NTZ) cell vs. WT or NC cells). This allows us to validate for the first time that NTZ, the only FDA-approved drug to treat human cryptosporidial infection, kills *C. parvum* by solely acting on the parasite target (100% on-target).

NTZ is a thiazolide compound with a relatively broad spectrum of activity against anaerobic bacteria and parasites by targeting pyruvate:ferredoxin/flavodoxin oxidoreductase (PFOR) involved in anaerobic metabolism. NTZ displays low micromolar inhibition constant ($K_i$ = 2 to 10 μM) on PFOR from the protozoan parasites *Trichomonas vaginalis*, *Entamoeba histolytica* and *Giardia intestinalis* and the bacterial pathogens *Clostridium difficile*, *C. perfringens*, *Helicobacter pylori* and *Campylobacter jejuni* [27]. However, the mode of action of NTZ against cryptosporidial infection is yet undefined. *Cryptosporidium* possesses a PFOR that is fused with an NADPH-cytochrome P450 reductase to form a unique bifunctional enzyme pyruvate:NADP$^+$ oxidoreductase (PNO) [28]. It should be noted that NTZ could also stimulate the host immune system [29–31]. This may partly explain why NTZ, which only displays lower micromolar in vitro anti-cryptosporidial efficacy (i.e., $EC_{50}$ from 1 to 5 μM as reported here and in other studies) [32,33], is only effective in immunocompetent patients. Nonetheless, the confirmation that NTZ is fully on-target justifies that the parasite PNO is worth to be investigated for developing more selective and effective anti-cryptosporidial drugs.

To our knowledge, this study represents the first attempt to develop algorithms for quantifying the proportional contributions of on-target and off-target effects to the overall anti-infective efficacy. Although the algorithms are developed under the experimental conditions used in this study, they are modifiable to suit other experimental settings. One noteworthy application of the algorithms is to evaluate and quantify the on/off-target effects in developing drugs targeting host cell targets localized inside the cells.

It is worth to clarify again that: 1) the *MDR1*-transfected cell models are applicable only to evaluating the on/off-target effects on epicellular pathogens (e.g., *C. parvum*) whose drug exposure would not be affected by MDR1 efflux. It is not applicable to pathogens residing in the host cytoplasm (e.g., *Toxoplasma* and *Eimeria* parasites) whose drug exposures would also be affected by MDR1 efflux; 2) The in vitro models and algorithms are used to evaluate whether an applicable inhibitor kills the parasite by acting fully or partially on the parasite target, rather than evaluating whether an inhibitor acts on a specific biochemical target; 3) The model can

only be used to evaluate whether a compound acts on the parasite target, but unable to identify or confirm the target. It also cannot evaluate drug's effect on immune response.

## Conclusions

We have developed an *MDR1*-transgenic cell-based model applicable to evaluating whether anti-cryptosporidial hits/leads kill the parasite by fully or partially targeting the parasite targets. The hits/leads can be either MDR1 substrates or non-MDR1 substrates. Using the model, we have validated that paclitaxel (PTX) and nitazoxanide (NTZ) kill *C. parvum* by fully acting on the parasite targets (100% on-target), while mitoxantrone (MTX), doxorubicin (DXR), vincristine (VCT) and ivermectin (IVM) kill the parasite by acting on both the parasite and host cell targets (partially on-target). We have also developed algorithms to quantify the percent contributions of on- and off-target effects to the observed anti-cryptosporidial efficacy in vitro, and to examine the relationships between anti-cryptosporidial efficacy ($EC_i$), cytotoxicity ($TC_i$), selectivity index ($SI$ or $k$) and Hill slope ($h$).

## Materials and methods

### *In vitro* culture of *C. parvum* and assays for anti-cryptosporidial efficacy and drug tolerance in host cell lines

A strain of *C. parvum* with subtype IIaA17G2R1 at the *gp60* locus was propagated in the laboratory by infecting calves, from which oocysts were collected from feces and stored in PBS containing 200 U/mL penicillin and 0.2 mg/mL streptomycin at 4˚C until use. Prior to use, oocysts were purified by a sucrose/CsCl gradient centrifugation protocols, followed by a 5-min treatment of 10% house bleach in ice and extended washes with distilled water [34–36]. The viability of the oocysts was assessed by in vitro excystation in PBS containing 0.5% taurocholic acid sodium salt hydrate at 37˚C for 1 h, and only those with >85% excystation rates were used in experiments.

HCT-8 cells (National Collection of Authenticated Cell Cultures, Shanghai, China) was used as a parent wild-type (WT) cell line for generating *MDR1*-transgenic cells for assaying in vitro drug efficacy against *C. parvum*. Host cells were maintained in 25 cm² flasks containing RPMI-1640 medium, 10% fetal bovine serum (FBS) and 1.0% penicillin-streptomycin at 37˚C under 5% $CO_2$ atmosphere. Anti-cryptosporidial efficacy assay was performed using an established protocol [37,38]. Briefly, host cells including WT and its derived transgenic cell lines were seeded in 96-well plates overnight until ~80% confluence and inoculated with *C. parvum* oocysts ($2\times10^4$ per well). After 3 h incubation to allow excystation and invasion of *C. parvum* sporozoites, uninvaded parasites were removed by a medium exchange. Compounds at specified concentrations were added at this time point, and infected host cells were incubated for additional 41 h (total 44 h of infection). Cell lysates were prepared using an iScript qRT-PCR sample preparation reagent (50 μL/well) (Bio-Rad Labs, California, CA) [37,38].

Cell lysates were diluted by 100-fold for detecting *C. parvum* 18S (*Cp18S*) rRNA and 2000-fold for detecting host cell 18S (*Hs18S*) rRNA by qRT-PCR using TransScript Green One-Step qRT-PCR SuperMix (TransGen Biotech, Beijing, China) in a StepOnePlus Real-Time PCR System (Applied Biosystems, Waltham, MA, USA). Each reaction included 3 μL diluted cell lysate, 10 μL 2× SuperMix solution, 0.4 μL forward and reverse primers (10 μM each), 0.4 μL Passive Reference Dye I, 0.4 μL TransScript One-Step RT/RI Enzyme Mix, and 5.4 μL RNase-free Water (total 20 μL), using primers specified in S4 Table. Anti-cryptosporidial activity was indicated by half-maximal effective concentration ($EC_{50}$ values) by nonlinear regression with a 4-parameter logistic (4PL) model using Prisms (v9.0; GraphPad, San Diego, CA).

Host cell drug tolerance to specified inhibitors was evaluated by 3-(4,5-dimethylthiazol-2-yl)-5-(3-carboxymethoxyphenyl)-2-(4-sulfophenyl)-2H-tetrazolium (MTS) assay. WT and transgenic host cells were seeded in 96-well plates (10,000 cells/well) and cultured for 24 h, followed by the addition of specified compounds at serially diluted concentrations and continued culture for 41 h. After 3 washes with serum-free medium, MTS solution (Saint-Bio, Shanghai, China) was added into the plates (20 μL/well) and incubated at 37°C for 2 h. Optical density at 490 nm ($OD_{490}$) was measured using a Synergy LX multi-mode reader (BioTek, Winooski, VT). Drug tolerance was indicated by half-maximal cytotoxic concentrations ($TC_{50}$ values) calculated by nonlinear regression using 4PL model. Selectivity index ($SI$) for each compound was determined by the ratio between $TC_{50}$ and $EC_{50}$ values ($SI = TC_{50}/EC_{50}$) [39,40].

## Development of stable transgenic cell lines and detection of *MDR1* gene expression

A lentiviral expression vector system including pCDH-CMV-MDR1-EF1α-copGFP-T2A-puro lentivector, psPAX2 and pMD2G helper plasmids was used to generate *MDR1*-transgenic cell lines (Xiamen Anti-hela Biological Technology Co., Xiamen, China). Blank vector pCDH-CMV-MCS-EF1α-copGFP-T2A-puro was used as negative control (Fig 2A). Recombinant lentiviruses were prepared by co-transfection of 293T cells with the vectors/plasmids, followed by collection of lentiviruses in the supernatant and determination of the viral titers [41]. Parent HCT-8 cells (WT) were infected with the lentiviral preparations for 48 h, followed by selection with puromycin (4 μg/ml) for 7 days. The resulting transgenic cell lines were designated as HCT-8/MDR1 (or MDR1 for short) that overexpressed MDR1 and HCT-8/NC (or NC) that carrying negative control blank vector (Table 1).

The morphology of WT, NC and MDR1 cells were examined by immunofluorescence assay (IFA), in which cells were cultured in 48-well plates containing glass coverslips coated with 0.1 mg/mL poly-*L*-lysin for 1 d. Cell monolayers were fixed in 4% paraformaldehyde for 10 min and permeabilized with 0.5% Triton X-100 in PBS for 5 min, followed by blocking with PBS buffer containing 3% BSA. MDR1 was detected by incubation with a rabbit monoclonal anti-MDR1 antibody (Cell Signaling Technology Co., Danvers, MA) (1:200 dilution) overnight at 4°C and anti-rabbit antibody conjugated with Alexa Fluor 594. There were three washes with PBS for 5 min after each treatment step. The same IFA procedures was also used to detect whether there was any enrichment of MDR1 at the host cell-parasite interface in specified host cells infected with *C. parvum* for 24 h.

The relative levels of *MDR1* expression were determined at protein and mRNA levels by western blot analysis and qRT-PCR, respectively. WT, NC and MDR1 cells were cultured in 24-well plates for 1 d or as specified and collected for preparation of protein extracts and isolation of total RNA. In western blot analysis, host cells were washed three times with PBS and lysed in radio-immunoprecipitation assay (RIPA) buffer (Sigma-Aldrich Co., Saint Louis, MO, USA; 50 μL/well). Proteins extracts (15 μg/lane) were fractionated by 10% SDS-PAGE and transferred onto nitrocellulose membranes. The blots were incubated with rabbit anti-MDR1 (1:1000) (Abcam, Cambridge, UK; Cat. # ab170904) or rabbit anti-GAPDH antibodies (1:5000) (Proteintech Inc., Rosemont, IL, USA; Cat. # 10494-1-AP) in PBS buffer containing 5% skim milk overnight at 4°C, followed by incubation with horseradish peroxidase (HRP)-conjugated Affinipure goat anti-rabbit IgG (1:5000) (Proteintech; Cat. # SA00001-2) for 1 h. The blots were developed using FGSuper Sensitive ECL Luminescence Reagent (Meilunbio, Dalian, China).

In qRT-PCR assay, total RNA was isolated from cells using Trizol RNA isolation kit (Takara, Shiga, Japan) and *MDR1* and *GAPDH* transcripts were detected using a TransScript

Green One-Step qRT-PCR SuperMix (TransGen Biotech, Beijing, China). The reactions (20 μL/reaction) contained 20 ng total RNA, 10 μL 2× SuperMix solution, 0.4 μL forward and reverse primers (10 μM each), 0.4 μL passive reference dye I, 0.4 μL TransScript One-Step RT/RI Enzyme Mix and 5.4 μL RNase-free water, and were performed using a StepOnePlus real-time PCR system (Applied Biosystems, Waltham, MA). Primers for *MDR1* and *GAPDH* were listed in S4 Table.

## Generation of cell lines with increased drug tolerance to MDR1 substrate paclitaxel (PTX) and non-substrate nitazoxanide (NTZ)

Stable *MDR1*-transgenic cell line (MDR1 cells) was more resistant than WT and NC cell lines to five of the nine compounds tested in this study, but the increases were less than 2-fold (ranging from 1.54 to 1.76) (Table 2), which were less ideal for evaluating on/off-target effects for these compounds and useless in evaluating other compounds. Since MDR1 was responsible for the development of multidrug resistance in cancer cells for a large number of therapeutics [42–44], we hypothesized that overexpression of *MDR1* would make host cells more adaptable than WT and NC cells to the drug selection pressure for rapid increase of resistance to MDR1 substrates (e.g., PTX) and induction of resistance to non-substrates (e.g., NTZ). To test the hypothesis, WT, NC and MDR1 cells were subjected to selection by PTX and NTZ.

We employed a drug selection scheme similar to those reported by other investigators [45–47], in which cells were subjected to multiple rounds of drug selection with incrementally increased drug concentrations, each round containing 2–3 cycles of 2-d drug selection at ~80% inhibition concentrations followed by 3–5 d of drug withdrawal to allow the growth of host cells to near confluence (see S2 Table for detailed drug selection design). More specifically, WT, NC and MDR1 cells were cultured in 6-well plates ($2\times10^5$ cells/well) to confluence and incubated with PTX at 0.75 μM (WT and NC cells) or 1.5 μM (MDR1 cells) or NTZ at 3.0 μM (WT, NC and MDR1 cells) for 2 d (the drug concentrations were near their $TC_{80}$ values determined by 48-h cytotoxicity assay). Surviving cells were allowed to recover in drug-free medium for 3–5 d to near confluence (round 1). The selection/recovery cycle were repeated once (round 2). Cells were then subjected to a serial new rounds of selection/recovery cycles with incrementally increased drug concentrations until MDR1 cells could grow normally in the presence of 7.61 μM of PTX or 15.20 μM of NTZ (round 11). At this time point, WT and NC cells could grow normally in the presence of 1.70 μM PTX or 10.13 μM NTZ (S2 Table). Finally, all cells were cultured at the final selection concentrations of PTX or NTZ for ≥7 d, followed by culture in drug-free medium for 14 d. At this time point, cells were used for cytotoxicity and efficacy assays or cryopreserved in a liquid nitrogen tank. The resulting cell lines after PTX or NTZ selection were designated as WT(PTX), NC(PTX) and MDR1(PTX), or WT(NTZ), NC(NTZ) and MDR1(NTZ), respectively (Tables 1 and S2).

## Mathematical models for quantitative estimation of relative contributions from the on- and off-target effects to the anti-cryptosporidial efficacy

**Model based on $EC_{50}$ and $TC_{50}$ values.** Let us denote $E_{on}$ and $E_{off}$ as the on- and off-target rates, and $E_{obs}$ as the observed as anti-parasitic efficacy, representing the proportions or precents of on/off-target effects contributing to the observed anti-cryptosporidial efficacy. The observed anti-parasitic efficacy ($E_{obs}$) is the sum of $E_{on}$ and $E_{off}$ that was set to 100%:

$$E_{obs} = E_{on} + E_{off} = 100\% \tag{6}$$

Under the condition that the drug tolerance is significantly increased in the drug-resistant cell line (e.g., >2-fold increase between $TC_{50(MDR1)}$ and $TC_{50(NC)}$), where MDR1 represents

MDR1-derived cell lines such as MDR1(PTX) and MDR1(NTZ) cells, the relative contributions of $E_{on}$ and $E_{off}$ to $E_{obs}$ can be indicated by whether, and how much, the anti-parasitic efficacy is also increased proportionally. More specifically, we may estimate the percent contribution of $E_{off}$ to $E_{obs}$ by calculating whether and how the relative increase of anti-parasitic efficacy ($RI_{EC50}$) is proportionally correlated to the relative increase of drug tolerance between ($RI_{TC50}$), or the ratio between $RI_{EC50}$ and $RI_{TC50}$ using the following equations:

$$RI_{EC50} = \frac{\Delta EC_{50}}{EC_{50(NC)}} = \frac{EC_{50(MDR1)} - EC_{50(NC)}}{EC_{50(NC)}} = \frac{EC_{50(MDR1)}}{EC_{50(NC)}} - 1 \tag{7}$$

$$RI_{TC50} = \frac{\Delta TC_{50}}{TC_{50(NC)}} = \frac{TC_{50(MDR1)} - TC_{50(NC)}}{TC_{50(NC)}} = \frac{TC_{50(MDR1)}}{TC_{50(NC)}} - 1 \tag{8}$$

$$E_{50(off)} = \frac{RI_{EC50}}{RI_{TC50}} = \frac{\left(\frac{EC_{50(MDR1)}}{EC_{50(NC)}} - 1\right)}{\left(\frac{TC_{50(MDR1)}}{TC_{50(NC)}} - 1\right)} \times (100\%) \tag{9}$$

Eq 9 can be rearranged to obtain Eq 1 described in the Results section. Based on Eq 6, we also obtain Eq 2 described in the Results section.

**Expansion of the model to the whole efficacy range from $EC_0$ to $EC_{100}$.** Dose-dependent efficacy and cytotoxicity kinetic curves generally follow a 4-parameter logistic (4PL) sigmoidal model [48]:

$$Y = \frac{E_{max} - E_{min}}{1 + \left(\frac{E_{50}}{X}\right)^h} + E_{min} \tag{10}$$

where $Y$ is the response (theoretically ranging from 0 to 1 probability values) and $X$ is the drug concentration. $E_{min}$ and $E_{max}$ are the lower and upper plateaus of the curve (also termed Bottom and Top). The parameter $h$ is the slope factor of the curve (Hill slope). The $E_{50}$ (= either $EC_{50}$ or $TC_{50}$) is the concentration to achieve the midway response between $E_{min}$ and $E_{max}$.

In a drug efficacy assay based on quantitation of relative parasite loads by qRT-PCR and a cytotoxicity test based on colorimetric or fluorescent assay, the response ($Y$) can be converted to the percent inhibition on the parasite or on host cell, in which $E_{min}$ is normalized to zero (i.e., the response to diluent in the negative controls). Eq 10 is then simplified to:

$$Y = \frac{E_{max}}{1 + \left(\frac{E_{50}}{X}\right)^h} = \frac{E_{max} \cdot X^h}{E_{50}{}^h + X^h} \tag{11}$$

Ideally, the parameter $E_{max}$ value is 1 (100%), by which $E_{50}$ (solved from the equation) is the inhibitor's concentration that truly achieves 50% inhibition, referred to as "absolute $EC_{50}$ or $TC_{50}$" [48]. However, $E_{max}$ might not reach 100% in many assays, in which $E_{50}$ solved from Eq 10 is relative to the upper plateau, referred to as "relative $EC_{50}$ or $TC_{50}$" (Note: this study reported relative $EC_{50}$ or $TC_{50}$ values).

**Derivation of equations to visualize the relationship between selectivity index (SI) and cytotoxicity over a drug's efficacious concentrations.** The principal here is to plot the cytotoxicity (inhibition rates on host cell growth; denoted by $Y_{TC}$) of a specified inhibitor against the concentrations of the inhibitor over the range showing anti-cryptosporidial efficacy

(denoted by $Y_{EC}$) in WT cells. Based on the 4PL model (Eq 10), we have:

$$Y_{TC(ECi)} = \frac{X_{ECi}{}^h}{TC_{50}{}^h + X_{ECi}{}^h}$$ (12)

where $X_{ECi}$ is the concentration of the inhibitor at anti-cryptosporidial efficacy *EC*. $Y_{TC(Ei)}$ is the cytotoxicity rate of the inhibitor at the concentration $EC_i$ ($i = 0$ to 100%).

Since SI is defined by the ratio between $TC_{50}$ and $EC_{50}$, we have:

$$TC_{50} = SI \times EC_{50} = k \times EC_{50}$$ (13)

where *k* is *SI* for simplicity. The parameter *k* in Eq 13 can be introduced into Eq 12:

$$Y_{TC(ECi)} = \frac{X_{ECi}{}^h}{(k \times EC_{50})^h + X_{ECi}{}^h}$$ (14)

The anti-cryptosporidial efficacy, denoted by $Y_{ECi}$ here for clarity, can be introduced into Eq 14 to replace $X_{ECi}$ based on the 4PL model again:

$$Y_{ECi} = \frac{X_{ECi}{}^h}{EC_{50}{}^h + X_{ECi}{}^h}$$ (15)

which can be derived to:

$$X_{ECi}{}^h = \frac{Y_{ECi} \times EC_{50}{}^h}{1 - Y_{ECi}}$$ (16)

After placing Eq 16 into Eq 14 and some derivations, we obtain the following simplified equation to define $Y_{TC(ECi)}$ as the function of $Y_{ECi}$, *k* and *h*, i.e., Eq 5 in the Results section.

In Eq 5, the *h* values in both efficacy and cytotoxicity curves are assumed to be the same after considering that a specified inhibitor would likely act on the same or similar targets in the parasite and the host cells. This assumption is also supported by the *h* values for the six inhibitors obtained in this study, in which the *h* values range from 1.0 to 1.24 and differ by 0.40% to 2.87% between efficacy and cytotoxicity curve pairs (S6 Table).

## Data analysis and statistics

At least two independent experiments were conducted for each experiment condition. Each experiment contained minimal 2 or biological replicates for experimental groups or negative controls, respectively. In qRT-PCR assay used 2 or 3 technical replicates. In vitro efficacy and cytotoxicity data were analyzed using Prism (v9.0 or higher; GraphPad, San Diego, CA) using a 4-parameter logistic model. Statistical significances were evaluated by two-way analysis of variance (ANOVA) and Holm-Šídák multiple *t*-test between group pairs [40].

## Supporting information

**S1 Fig. Illustration of the possible actions of inhibitors on *Cryptosporidium parvum* in vitro. (A)** Diagram of a developing meront of *C. parvum* in vitro. This epicellular parasite is contained within a parasitophorous vacuole membrane (PVM) derived from host cell plasma membrane (thus intracellular), but separated from host cell cytoplasm by an electron-dense (ED) layer (thus extra-cytoplasmic). **(B)** The observed anti-cryptosporidial efficacy of an inhibitor could be attributed to: 1) the action solely on the parasite target (= fully on-target); 2) the action soley on the host cell target (= fully off-target); or 3) on both the parasite and host cell targets (= partially on-target). Depending on the property of the inhibitor, on- and off-

target effects might contribute to the observed anti-cryptosporidial activity at varied levels.
(TIF)

**S1 Table. List of the compounds used in this study and effect of overexpression of MDR1 and drug selection by PTX or NTZ on drug tolerance profiles.**
(XLSX)

**S2 Table. Drug selection strategy and experimental design.**
(XLSX)

**S3 Table. Relationship between anti-cryptosporidial efficacy ($EC_{50}$), drug tolerance as indicated by cytotoxicity ($TC_{50}$) and selectivity index ($SI$) of selected compounds in specified cell lines.**
(XLSX)

**S4 Table. PCR primers used in this study.**
(XLSX)

**S5 Table. Percent contributions of the on-target and off-target effects of the six compounds to the anti-cryptosporidial activity in vitro at selected EC values (between $EC_{10}$ and $EC_{90}$) as calculated using Eqs 3 and 4.**
(XLSX)

**S6 Table. Hill slope (h) values in the anti-cryptosporidial efficacy and cytotoxicity assays in all cell lines.**
(XLSX)

## Author Contributions

**Conceptualization:** Guan Zhu.

**Data curation:** Bo Yang, Guan Zhu.

**Formal analysis:** Bo Yang, Yueyang Yan, Guan Zhu.

**Funding acquisition:** Jigang Yin, Guan Zhu.

**Investigation:** Bo Yang, Yueyang Yan, Dongqiang Wang, Ying Zhang, Jigang Yin.

**Methodology:** Bo Yang, Dongqiang Wang, Guan Zhu.

**Project administration:** Guan Zhu.

**Software:** Yueyang Yan.

**Supervision:** Guan Zhu.

**Validation:** Bo Yang, Guan Zhu.

**Visualization:** Bo Yang, Yueyang Yan, Guan Zhu.

**Writing – original draft:** Bo Yang, Guan Zhu.

**Writing – review & editing:** Guan Zhu.

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
