## [Decision Letter · Decision Letter 0]

20 Sep 2022

Dear Dr. Zhu,

Thank you very much for submitting your manuscript "On-target inhibition of Cryptosporidium parvum by nitazoxanide (NTZ) and paclitaxel (PTX) validated using a novel MDR1-transgenic host cell model and algorithms to quantify on/off-target rates" for consideration at PLOS Neglected Tropical Diseases. As with all papers reviewed by the journal, your manuscript was reviewed by members of the editorial board and by several independent reviewers. In light of the reviews (below this email), we would like to invite the resubmission of a significantly-revised version that takes into account the reviewers' comments. 

Reviewer's Comments:

1. The authors developed a novel anti-cryptosporidial drug model in vitro. How to evaluate the efficacy of these drugs anti-cryptosporidial effects (kill Cryptosporidium)? The methods in this manuscript was limited, which could not effectively evaluate the anti-cryptosporidial effects. So, these results and limited should be describe in discussion part.

2. In the cytotoxicity test, the biological function changes of HCT-8 cells should be added.

3. The function of MDR1 and it’s function in Cryptosporidium infection should be described in background.

4. According to the methods and the results in this study, the authors validated that NTZ killed C. parvum by solely acting on the parasite target. There was limited.

5. Line 3: Use of the terms on-target and off-target are confusing here, and not aligned with their standard use in pharmacology. “On-target” typically refers to action on the target of the therapeutic effect and “off-target” usually refers to a toxic effect. Instead, consider “parasite-target” and “host-target”.

6. Line 3: Use of the term “rates” in the title (“on/off-target rates”) is confusing, as this term typically refers to something happening over time. It also evokes an “off-rate” in a molecular binding, which is not at all what is being referred to here. I suspect that what the authors mean is “ratio”, but I suggest using the term “effect”, as in the Short Title.

7. Lines 95-100: I suggest an additional comment here about why it is important to understand MOA. For example, better understanding of MOA can enable optimization of drugs, reduction in toxic effects, and identification of new leads with the same or similar MOA.

8. Line 134: A critical hypothesis that is not at all addressed in this manuscript is that NTZ’s broad spectrum antimicrobial activity (reported against bacteria, viruses, and parasites) is due to stimulation of the host immune system, rather than a direct effect on pathogens (Jaseonsky et al. 2019 iScience DOI: 10.1016/j.isci.2019.07.003; Dang et al. 2018 Antimicrob Agents Chemother DOI: 10.1128/AAC.00707-18; Elazar et al. 2009 Gastroenterol DOI: 10.1053/j.gastro.2009.07.056; etc.). While this hypothesis does not exclude the possibility that NTZ may have direct effects on Cryptosporidium and/or direct effects on infected host cells that inhibit parasite growth, it should be acknowledged that this HCT-8/MDR1-transgenic model does not enable investigation of NTZ effects on immune response.

9. Line 155: drugs are “approved” or “registered” by the US FDA, only vaccines are “licensed”

10. Line 184: “…develop drug resistance more rapidly…”? According to Table S2, all lines were selected for the same duration. Instead consider, “…develop greater drug resistance…”

11. Lines 209-210: If there is no increase in MDR1 protein expression by IFA, what is the basis of increased PXT or NTZ tolerance in selected cells?

12. Line 360: Implying the ratio of TC50/EC50 is a major determinant of clinical drug safety could be considered miseading. Dose-limiting drug toxicity is usually not driven by cytotoxicity directly, but instead by toxic effects on other tissues (heart/QT, liver/metabolic pathways, etc.). Consider using the term “in vitro cytotoxicity index” rather than “safety interval”.

13. Lines 431-441: As mentioned previously above, there should be some discussion of the potential impact of NTZ on immune modulation.

14 Line 539: Method of quantitation of western blots (phosphoimager?) in Figs 2C and 2F should be described. The bands on the westerns in Figs 2B and 2E look more than 1.5x darker/bigger.

15. Line 44: “…newly hits/leads” should be modified to “new hits/leads” or “newly identified hits/leads”.

16. Lines 70-71: For “…big weight loss”, consider instead “significant” or “substantial”.

17. Lines 83-84: …a parasite target…a host cell target… (indefinite articles are missing for both).

18. Line 132: inhibition (not “inhibitions”)

19. Line 134: delete “for” in “This is for the first time…”

20 Line 139: allow (not “allows”)

21. Line 143: promoters

22. Line 154: It is unnecessary to spell out name of compound/assay in Results. I suggest describing this detail only in the Materials and Methods section.

23. Figure 2A: pCMV, not “pMCV”

24. Figure 7A: “inhibior” is misspelled

25. Figure 11B: “efficay” is misspelled

We cannot make any decision about publication until we have seen the revised manuscript and your response to the reviewers' comments. Your revised manuscript is also likely to be sent to reviewers for further evaluation.

Sincerely,

Hamed Kalani

Academic Editor

S Madison-Antenucci

Section Editor
---

## [Editor Report · Decision Letter 1]

5 Mar 2023

Dear Dr. Zhu,

We are pleased to inform you that your manuscript 'On-target inhibition of Cryptosporidium parvum by nitazoxanide (NTZ) and paclitaxel (PTX) validated using a novel MDR1-transgenic host cell model and algorithms to quantify the effect on the parasite target' has been provisionally accepted for publication in PLOS Neglected Tropical Diseases.

Best regards,

Hamed Kalani

Academic Editor

Susan Madison-Antenucci

Academic Editor

---

## [Editor Report · Acceptance letter]

17 Mar 2023

Dear Dr. Zhu,

We are delighted to inform you that your manuscript, "On-target inhibition of Cryptosporidium parvum by nitazoxanide (NTZ) and paclitaxel (PTX) validated using a novel MDR1-transgenic host cell model and algorithms to quantify the effect on the parasite target," has been formally accepted for publication in PLOS Neglected Tropical Diseases.

Best regards,

Shaden Kamhawi

co-Editor-in-Chief

Paul Brindley

co-Editor-in-Chief
